# Low-temperature hydroformylation of ethylene by phosphorous stabilized Rh sites in a one-pot synthesized Rh-(O)-P-MFI zeolite

Minjie Zhao[1], Chengeng Li [1,2], Daviel Gómez [1], Francisco Gonell [1], Vlad Martin Diaconescu[3], Laura Simonelli [3], Miguel Lopez Haro [4], Jose Juan Calvino [4], Debora Motta Meira [5,6], Patricia Concepción [1] ✉ & Avelino Corma [1] ✉

Zeolites containing Rh single sites stabilized by phosphorous were prepared through a one-pot synthesis method and are shown to have superior activity and selectivity for ethylene hydroformylation at low temperature (50 °C). Catalytic activity is ascribed to confined $Rh_2O_3$ clusters in the zeolite which evolve under reaction conditions into single $Rh^{3+}$ sites. These $Rh^{3+}$ sites are effectively stabilized in a Rh-(O)-P structure by using tetraethylphosphonium hydroxide as a template, which generates in situ phosphate species after $H_2$ activation. In contrast to $Rh_2O_3$, confined $Rh^0$ clusters appear less active in propanal production and ultimately transform into $Rh(I)(CO)_2$ under similar reaction conditions. As a result, we show that it is possible to reduce the temperature of ethylene hydroformylation with a solid catalyst down to 50 °C, with good activity and high selectivity, by controlling the electronic and morphological properties of Rh species and the reaction conditions.

Controlling the structure of active sites at the atomic scale and understanding their dynamic rearrangement under reaction conditions is crucial in the design of efficient catalysts. Thus continuous effort has been directed to the development of efficient synthesis methods allowing the stabilization of metal sites with controlled oxidation state and coordination degrees, and in establishing structure-activity relationships. Sub-nanometric metal clusters and single metal sites offer a great opportunity to tune catalytic activity and selectivity, bridging the gap between homogeneous and heterogeneous catalysts, while modulating their local environment, metal-support interaction and electronic structure[1]. However, decoding active structures at the atomic level remains a significant challenge due to dynamic structural rearrangement of the catalyst under reaction conditions, requiring advanced nano and macro-scale characterization tools with sufficient

chemical and time resolution[2,3]. Additionally, because each technique has its own inherent limitations, complementary techniques are very necessary to prove the structure of active sites[3,4]. Regarding the synthesis of metal clusters or single metal sites, different approaches such as covalent bonding via strong metal-support interactions[5–7], immobilization methods in anionic or cationic supports such as zeolites or inorganic clays[8], and the use of ionic liquids have been considered[9]. Among them, the use of zeolites for stabilization of single metal sites and clusters of controlled atomicity under harsh conditions are of fundamental and industrial interest, as has been shown in recent reports dealing with zeolite confined Pt-, Pd-, Rh- and Ru-based clusters or single atom catalysts[10–18]. In addition, the low coordination degree of cations in zeolites and their high electrophilicity as compared to the same cation in other supports, allows for high

[1]Instituto de Tecnología Química, Universitat Politècnica de València-Consejo Superior de Investigaciones Científicas (UPV-CSIC), Avenida de los Naranjos s/n, 46022 Valencia, Spain. [2]Beijing Advanced Innovation Center for Soft Matter Science and Engineering, Beijing University of Chemical Technology, 100029 Beijing, P. R. China. [3]CELLS - ALBA Synchrotron Radiation Facility, Carrer de la Llum 2-26, 08290 Cerdanyola del Vallès, Spain. [4]Departamento de Ciencia de los Materiales e Ingeniería Metalúrgica y Química Inorgánica. Facultad Ciencias, Universidad de Cádiz, Campus Rio San Pedro, Puerto Real 11510-Cádiz, Spain. [5]Debora CLS@APS, Advanced Photon Source, Argonne National Laboratory, 9700 South Cass Avenue, Lemont, Illinois 60439, USA. [6]Canadian Light Source Inc., 44 Innovation Boulevard, Saskatoon, Saskatchewan S7N 2V3, Canada. ✉e-mail: pconcepc@upvnet.upv.es; acorma@itq.upv.es

coordination flexibility, tuning the Lewis acid properties of the metal cation. This is of particular importance in reactions involving intermediates with multiple coordinate ligands, as in the case of hydroformylation reactions. However, despite the protective role of zeolites in the stabilization of metal species, leaching of metal species into the solution in the case of liquid phase reactions or sintering of metal species under gas phase reaction conditions cannot be completely avoided, being strongly dependent on the reaction conditions. Hence, besides determining the structure of active sites and the corresponding catalytic behavior, a crucial step in the design of robust heterogeneous catalysts is outlining the factors that could stabilize the desired active species.

Hydroformylation can be seen as an example of a chemical process greatly affected by the local environment of active sites. It is one of the most important large-scale industrial processes to produce aldehydes, for which chemo- and regio-selectivity as well as catalyst stability are important objectives[19,20]. Industrially it operates using homogeneous organometallic catalysts based on single metal sites interacting with different types of phosphite and phosphine ligands[21,22]. Among the catalysts explored in this reaction, Rh catalysts are the most active ones, ranging from molecular Rh organocomplexes to solid supported metal nanoparticles and single sites[23–28]. Although the critical role of phosphine ligands in enhancing catalytic activity has been extensively discussed[29–31], their severe toxicity makes the development of alternative catalysts necessary. In this line, single-site solid-based catalysts have gained interest integrating the monoatomicity of organometallic homogeneous complexes with the advantages of heterogeneous processes. Moreover, the possibility of controlling the activity and stability of single metal sites by tuning their chemical properties and local environment opens expectations for catalytic applications, where zeolites offer an interesting structural environment for the stabilization of low coordinated metal sites. In this respect, for example, Rh@Y[32] and K-Rh@S-1 catalysts[33] have shown promising activity in the liquid phase 1-hexene and gas phase propylene hydroformylation, respectively. However, the nature of active sites has been controversially reported in the literature[34–37], probably due to the dynamic rearrangement of the metal sites under reaction conditions combined with the heterogeneity of metal species, making a set of multiple operando spectroscopic tools crucial for catalyst understanding. Moreover, the development of synthesis methodologies to make the Rh active species stable enough during operation by avoiding the use of phosphine capping agents is a key issue.

Here a novel synthesis strategy allowing the stabilization under hydroformylation reaction conditions of active Rh species in the absence of phosphine ligands is presented. In the first part of the work, a detailed analysis of the nature of active Rh species, their evolution and stabilization under hydroformylation working conditions is performed, using a Rh-MFI zeolite prepared by a one-pot hydrothermal method and submitted to different thermal treatments. Operando infrared spectroscopy coupled with mass spectrometry (IR-MS) is combined with high-angle annular dark field imaging and scanning transmission electron microscopy (HAADF-STEM), X-ray adsorption spectroscopy (XAS) and Infrared spectroscopy of CO as probe molecule (IR-CO) and validated with kinetic and catalytic studies in order to track the dynamic evolution of the catalytic active species. Thus, it is shown that the initial $Rh^0$ and $Rh_2O_3$ clusters, present in the reduced ($Rh^0$@MFI) and calcined ($Rh_2O_3$@MFI) catalysts respectively, disrupt under reaction conditions in different oxidized single site Rh species, $Rh(I)(CO)_2$ and high oxidation state $Rh^{3+}$, with the last one acting as a more effective precursor for low-temperature ethylene hydroformylation with syngas to produce propanal. The resultant Rh-MFI calcined (Rh-MFI-cal) catalyst is more active compared to state-of-the-art phosphine-free solid catalysts when operating at low temperature (50–100 °C), with TOF of 99 h$^{-1}$ at 90 °C and ~92% selectivity to the aldehyde. In addition to these results, in the second part of our work,

we present the possibility of stabilizing single $Rh^{3+}$ active site under reaction conditions, even under energetically favored metal sintering conditions, by developing a novel synthesis strategy in which phosphorous is introduced within the zeolite channels by using tetraethylphosphonium hydroxide as template. After $H_2$ activation it generates in situ phosphate species, stabilizing $Rh^{3+}$ sites and promoting propanal formation. The high steric hindrance of the zeolite channels promotes regioselectivity when using propylene as substrate, opening new perspectives in the design of regio-selective catalysts.

## Results

### Rh clusters stabilized in a one-pot synthesized MFI zeolite

A Rh-MFI catalyst containing 0.23 wt% Rh is prepared by a one-pot synthesis method using tetrapropylammonium hydroxide (TPAOH) as template, followed by air calcination at 550 °C (see more details of the synthesis in the Experimental section of Supplementary Information 3.1). Integrated differential phase contrast imaging (iDPC) combined with HAADF-STEM images, show the presence of sub-nanometric Rh clusters with particle sizes ~0.6–1.0 nm, selectively located within the sinusoidal channel of MFI structure (Fig. 1a, b and Supplementary Fig. S2). The oxidation state of the Rh species in this sample is determined by X-ray Adsorption Near-Edge Spectroscopy (XANES) and IR studies of CO adsorption at −65 °C. The sample XANES spectrum at the Rh K-edge (Fig. 1c, Supplementary Fig. S12) shows the rising edge overlapping that of $Rh_2O_3$, indicating a 3+ average oxidation state for the Rh centers. Furthermore, FT of the EXAFS data shows the dominant presence of Rh-C/N/O scattering in the 1−2 Å range, which nicely overlaps with the Rh-O contribution of the $Rh_2O_3$ reference (Fig. 1d). Moreover, a higher shell feature appears in the 2.2–3.1 Å range in correspondence to the Rh-Rh scattering contribution of the $Rh_2O_3$ system, but with a reduced intensity corresponding to a smaller, more disordered clustering, i.e., smaller particle size (Supplementary Table S4). Globally, the XANES and EXAFS data are compatible with the formation of small sub-nanometric $Rh_2O_3$ clusters. This result is supported by IR-CO data, which shows the presence of oxidized Rh species, comprising $Rh^{3+}$ (2154−2135 cm$^{-1}$)[38,39], $Rh^{2+}$ (2125 cm$^{-1}$)[40,41] and $Rh^+$ (2112 and 2030 cm$^{-1}$)[25,41,42] (Fig. 1e).

When Rh-MFI-cal zeolite is treated under hydroformylation reaction conditions with syngas, i.e., $C_2^-$/CO/$H_2$/$N_2$ (1/1/1/0.5 molar ratio), at 50 °C and 10 bar, within the spectroscopic XAS catalytic cell, both the XANES and EXAFS spectral region show a clear evolution as can be seen in Fig. 2a and b, respectively. A drop in white line intensity at 23240 eV accompanied by a decrease in the first shell coordination from FT EXAFS and a decrease in the Rh-Rh scattering at 2.8 Å indicate the decomposition of the $Rh_2O_3$ phase. In parallel, the shoulder around 23221 eV in the XANES and the peaks at 1.2 and 2.5 Å in the FT spectra, which become resolved during the reaction, suggest the formation of Rh-CO species[43], compatible with the operando IR data discussed later. Indeed EXAFS analysis using collinear Rh-CO scattering paths captures the formation of Rh-CO along with the formation of a minor amount of small Rh(0) clusters along the reaction coordinate (Supplementary Table S5). In addition, the contribution at 1.5 Å (Rh-O shell) in the FT also corresponds to oxidized Rh species (Fig. 2). Important, however, is that the rising edge of the evolving species overlaps that of $Rh_2O_3$, indicating the dominant presence of Rh centers having a high effective oxidation state. However, while the exact oxidation state of the Rh species cannot be defined by XAS because of the nature of the coexisting phases and the electron-withdrawing effects of coordinating CO ligands, IR-CO data (Supplementary Fig. S22), does however show the co-existence of $Rh^{3+}$ species (IR bands at 2154−2135 cm$^{-1}$) in the resting state of the catalyst after in situ exposed to reaction conditions.

Next to the XAS studies at 50 °C, raising the reaction temperature to 90 °C results in a slight increase of small Rh(0) clusters (details in Supplementary Information, Supplementary Fig. S16).

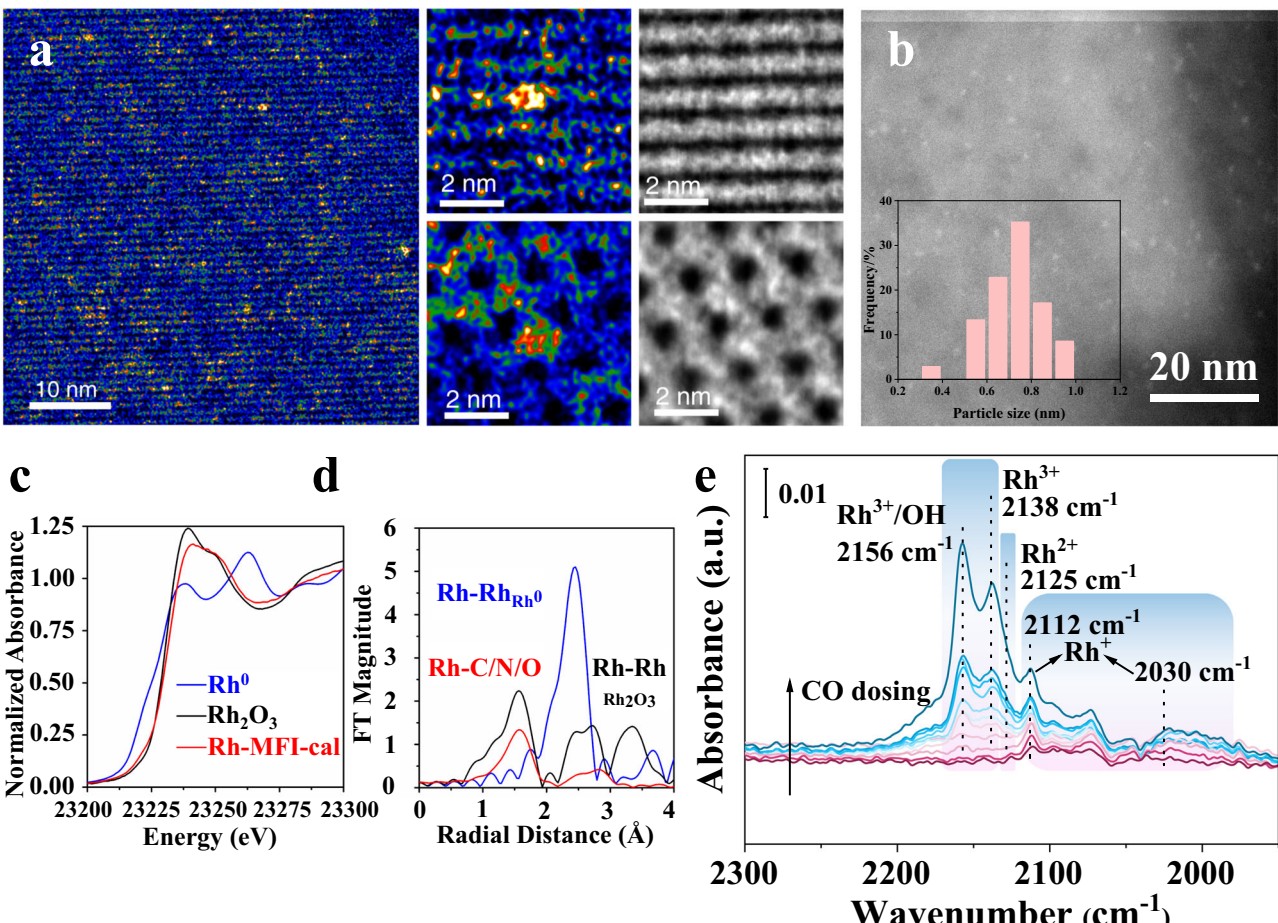

**Fig. 1 | Image and spectroscopic characterizations of Rh-MFI-cal zeolite.** Electron microscopic characterization of Rh-MFI-cal sample: **a** Paired HAADF-STEM and iDPC images and **b** HAADF-STEM image and particle size distribution. Rh clusters appear as small bright particles with particle sizes ~0.6–1.0 nm. **c, d** X-ray adsorption spectroscopic characterization of Rh clusters: **c** XANES spectra collected at the Rh K-edge (left) and **d** the Fourier Transform of the $k^2$-weigthed EXAFS function signal for Rh-MFI-cal and references. **e** IR-CO at −65 °C and at increasing CO coverage (0.01–2 mbar) for the Rh-MFI-cal sample. The contribution of CO coordinated to silanol groups (2156 cm$^{-1}$) and physisorbed CO (2135 cm$^{-1}$) to the Rh$^{3+}$-CO band is minimal, as determined from a blank experiment done on pure MFI (see Supplementary Fig. S24).

The used sample processed under the above-described reaction conditions, i.e., 50–90 °C and 10 bar in the presence of the $C_2^=$/CO/H$_2$/N$_2$ (1/1/1/0.5 molar ratio) gas mixture is also studied by STEM. In this case, the HAADF-STEM image of the Rh-MFI-cal after being submitted to hydroformylation reaction conditions in the 50–90 °C temperature range shows a very large proportion of isolated single Rh atoms (Supplementary Figs. S3 and S4b) with minor metal aggregates, revealing a disruption of the original ~0.6–1.0 nm Rh$_2$O$_3$ clusters into mainly single sites under reaction conditions. When the hydroformylation of $C_2^=$ is then carried out in a fixed-bed reactor under diffusion-free reaction conditions (see details in Supplementary Information 3.1), with a space velocity (GSHV) of 8000 h$^{-1}$ and under the reaction conditions described above, the catalyst is active and selective already at 50 °C. The yield to propanal is around 0.7, 1.6 and 2.2 mmol$_{propanal}$/g$_{cat}$.h., at 50, 70 and 90 °C respectively, and the selectivity is above 95% (Fig. 3a) (more information in Supplementary Information), being ethane production negligible (0.02, 0.03, 0.07 mmol$_{Ethane}$/g$_{cat}$.h at the above temperatures respectively). At 50 °C reaction conditions, the propanal production is stable, at least during the time of the experiment (160 min), and a slight catalyst deactivation is observed at 70 and 90 °C.

It is worth noting that the turnover frequencies (TOFs) for hydroformylation at 50, 70 and 90 °C are 31, 71 and 99 h$^{-1}$, respectively, which are sensibly higher than those reported in the literature for phosphine-free solid catalysts in the hydroformylation of ethylene (see Supplementary Tables S2 and S3 and Fig. 3b). It is worth mentioning that in most of the literature studies, temperatures above 100 °C are usually considered, probably due to the low catalyst activity at lower temperatures (see Supplementary Table S2). Furthermore, the apparent activation energy for our Rh-MFI single-site catalyst is much lower than the values reported in the literature for Rh supported catalyst (19.5 vs 50–70 KJ/mol) obtained in a fixed-bed reactor (Supplementary Fig. S7)[28,42]. Interestingly, a similar value of 20 KJ/mol has been recently reported by Christopher et al. for a 0.23 wt% Rh-0.7 wt% WO$_x$ /Al$_2$O$_3$ catalyst containing Rh$^{3+}$ sites[28]. Catalytic data are supported by operando IR-MS studies performed under relevant reaction conditions, where propanal (m/z = 58) is identified in the mass spectra at already 50 °C and 10 bar with almost 100% selectivity from 50 °C up to 90 °C reaction temperature (Supplementary Figs. S17 and S25). Interestingly, in the operando IR spectra, an intermediate monocarbonyl Rh(CO)L (L = propionyl) complex interacting with OH groups of the zeolite and characterized by a ν(Rh-CO) IR band at 2080 cm$^{-1}$ and a ν(C = O) of the ligand (L = CH$_3$CH$_2$CO*) at 1700 cm$^{-1}$, is identified under working conditions (details in Supplementary Information), being compatible with the aforementioned XANES data. This intermediate species is unstable in the absence of reactants and could only be detected under IR transient conditions (Supplementary Fig. S19). Indeed, when switching from the reactant feed at 90 °C into a flow of inert gas it is

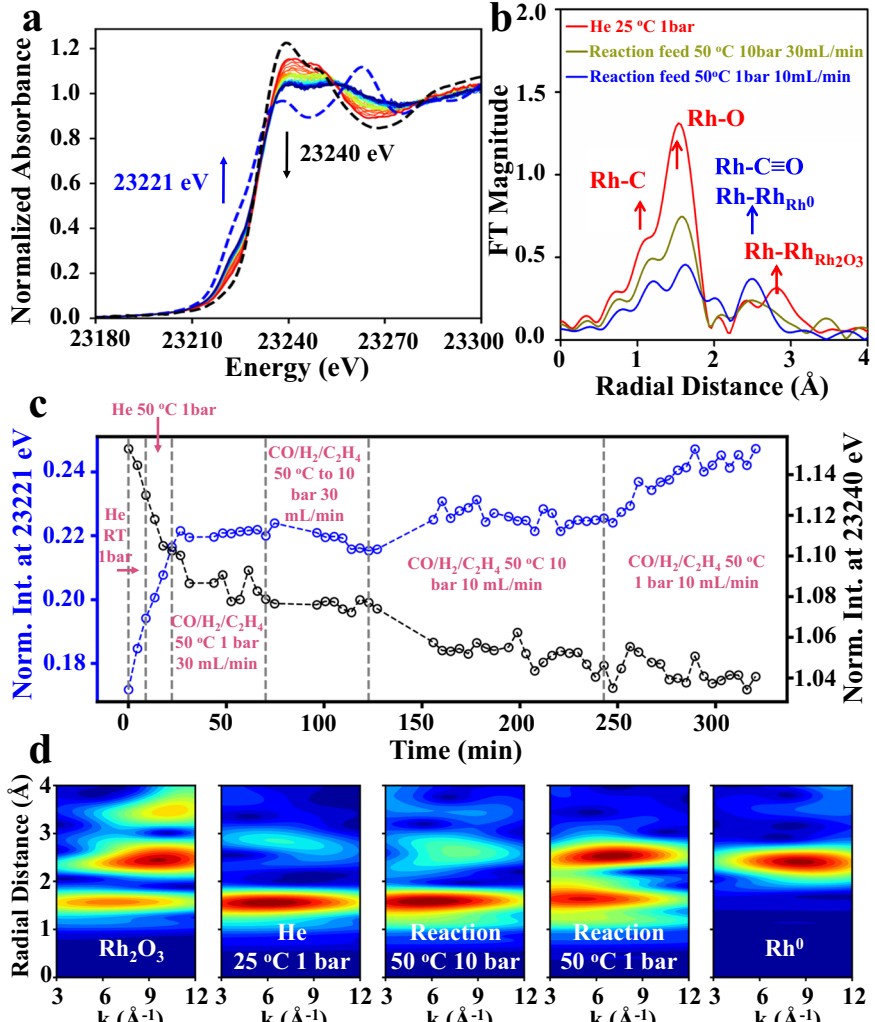

**Fig. 2 | Operando XAS studies of Rh-MFI-cal zeolite.** Evolution of the Rh K-edge XAS during operando studies in the Rh-MFI-cal sample. **a** Time-resolved XANES spectra at 50 °C, 10 bar and 30 mL/min reactant feed over a time interval of 250 min, (red to yellow and green) and after pressure decrease to 1 bar, 10 mL/min reactant feed and over a period of 100 min (blue lines). Spectra are overlaid with Rh(0) (blue dashes) and to $Rh_2O_3$ (black dashes) references; **b** Representative Fourier transformed $k^2$-weigthed EXAFS signal along the reaction coordinate. The arrows showing the different contributions are only indicative. The EXAFS fitting results are reported in the Supplementary Information; **c** Evolution of the normalized spectral intensities at 23221 eV corresponding to Rh-CO/Rh(0) (blue) and 23240 eV corresponding to $Rh_2O_3$ (black); **d** Cauchy wavelet transform of spectra in (**b**) showing evolution of peaks between 1.2 and 2.2 Å indicative of interference from Rh-CO and Rh-O scattering as well as small amounts of $Rh^0$ clustering in the final stages of the reaction (2.5 Å).

quickly transformed into $Rh(I)(CO)_2$ (IR bands at 2100–2090 and 2032 cm$^{-1}$), remaining some Rh as 3+ (see IR CO titration spectra at −65 °C in Supplementary Fig. S22). It is important to note that $Rh^{3+}$ is easily reduced to $Rh^+$ in the presence of CO, a behavior that is enhanced by the temperature and even happens at low temperature (details in Supplementary Information). This makes the detection and quantitative analysis of $Rh^{3+}$ species by IR-CO challenging. In conclusion, taking all the above spectroscopic and catalytic results into consideration, single oxidized Rh species in an effective high oxidation state (presumably as $Rh^{3+}$), generated by an in situ disruption of $Rh_2O_3$ clusters are proposed as an effective precursor for propanal formation.

This conclusion differs from what is commonly reported in the literature for heterogeneous catalysts, where low oxidized states such as $Rh^+$ in the form of $Rh(I)(CO)_2$ or $Rh^0$ are proposed as active sites in hydroformylation[24,34,44]. Interestingly, Wang et al.[45] in a motivating work have identified Rh in an oxidation state 3+ by XPS on their most active 0.2 wt% Rh-CoO catalyst, concluding that single site in a high oxidation state is essential for hydroformylation, but a detailed analysis of the impact of oxidation states was not done. Next, the catalytic

properties of $Rh(I)(CO)_2$ species, commonly accepted as the active site in heterogeneous hydroformylation catalysts are analyzed and contrasted to the one of this study. To do that, we carried out a controlled IR-MS experiment in which the Rh-MFI-cal catalyst was submitted to a syngas flow at 120 °C prior to the reaction, resulting in the formation of isolated $Rh(I)(CO)_2$ species (details in Supplementary Information). A comparative analysis of the MS pattern (Supplementary Fig. S29) shows negligible propanal formation at a temperature of 50 °C on the syngas pre-activated sample containing $Rh(I)(CO)_2$, with an onset temperature of 104 °C for propanal formation which is much higher than the one required for the calcined catalyst (i.e., 50 °C). The lower reactivity of $Rh(I)(CO)_2$ is in line with the dissociate hydroformylation reaction mechanism and the high stability of dicarbonyl $Rh^+$ species, where the removal of one CO ligand is required prior to olefin coordination[23,46]. The herein spectroscopic data obtained on the syngas pre-activated Rh-MFI sample are validated by catalytic data done in a fixed-bed reactor. Lower propanal yields (0.4, 0.6, and 1.4 mmol$_{propanal}$/g$_{cat}$.h at 50, 70 and 90 °C, respectively) (Supplementary Fig. S31), and a higher apparent activation energy for propanal

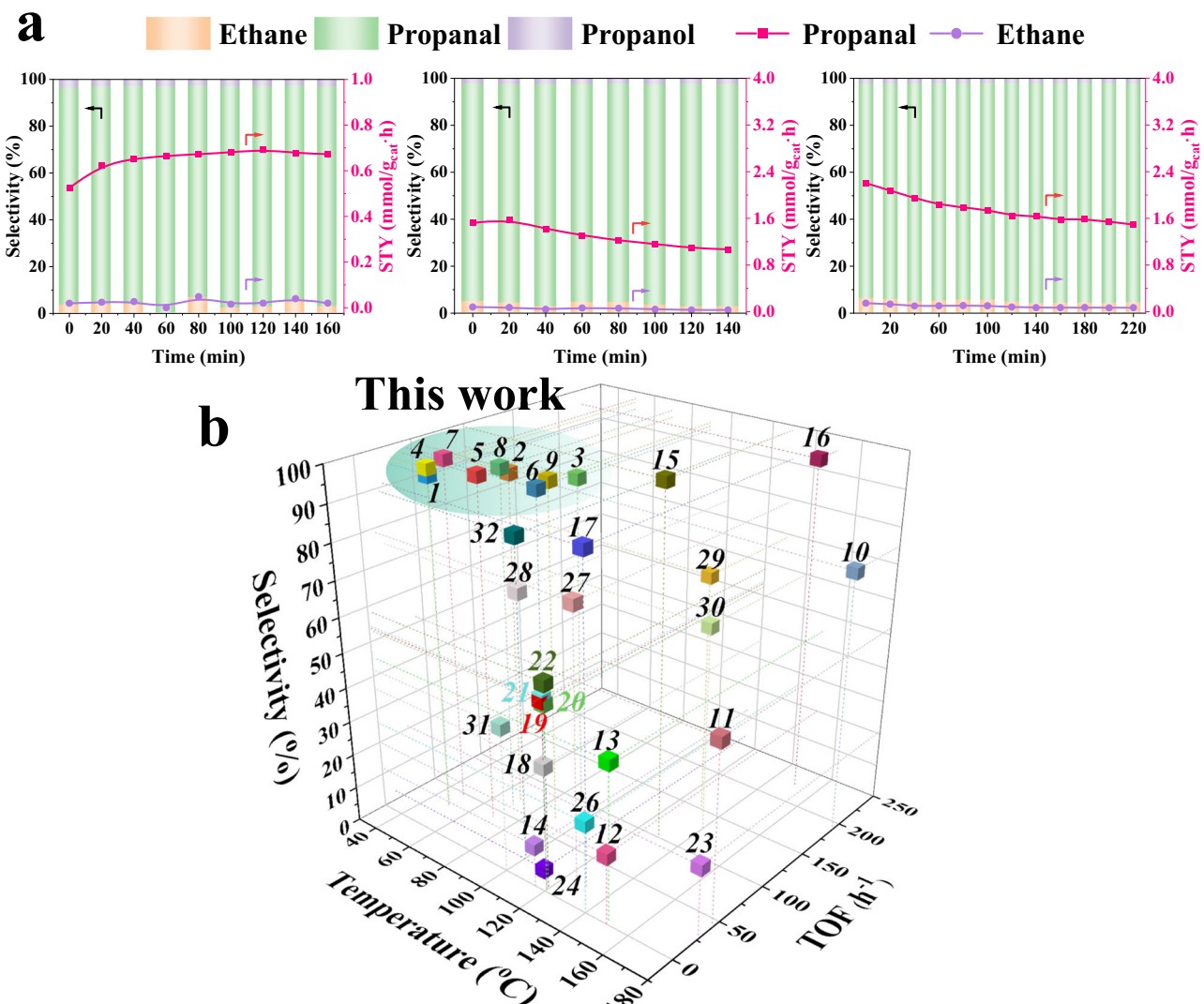

**Fig. 3 | Catalytic performance of Rh-MFI-cal zeolite in ethylene hydroformylation with syngas and comparison with state-of-the-art catalysts. a** Propanal (red line) and ethane (violet line) space time yield (STY) with corresponding values at the right axe, at 50, 70 and 90 °C (left, middle and right panels respectively), and selectivity to propanal (green bar), ethane (brown bar) and propanol (violet bar) with corresponding values on the left axe. **b** 3D map of TOF and propanal selectivity versus temperature of state-of-the-art-phosphine-free solid catalysts. Source data are provided as a Source Data file.

formation (25.6 KJ/mol) are observed on this sample compared to that of the non-activated sample (Supplementary Fig. S32). The observed differences between both samples might indicate a less favored process when starting from a pre-catalyst containing initially isolated Rh(I) $(CO)_2$ sites (i.e., syngas treated sample) rather than $Rh_2O_3$ clusters (i.e., non-treated sample). Moreover, selectivity around 92–95% is obtained in the syngas pre-activated sample, in parallel with the detection of predominantly single Rh sites (i.e., Rh(I)$(CO)_2$) after operando IR-MS hydroformylation reaction (see Supplementary Fig. S30).

In the second step, the performance of the Rh-MFI-cal catalyst containing initially $Rh_2O_3$ clusters is compared to that of the same sample containing $Rh^0$ clusters. With this purpose, the Rh-MFI-cal sample is reduced in $H_2$ flow at 600 °C (Rh-MFI-calred) resulting in the formation of Rh clusters with particle size 0.8–1.8 nm (Supplementary Figs. S33 and S34) located inside the sinusoidal channel of MFI framework, according to HAADF-STEM images. IR-CO of the reduced sample confirms the existence of predominantly $Rh^0$ clusters (2064 cm$^{-1}$)[41] and some Rh$^+$ (2104 and 2025 cm$^{-1}$) (Supplementary Fig. S35). Also, the XAS spectra confirm this result, with the spectrum of the Rh-MFI sample under $H_2$ reduction approaching that

of Rh(0) (Supplementary Fig. S40a). Nevertheless, when the reduced sample is treated under reaction conditions, ie $C_2^=/CO/H_2$ flow at 50 °C and 10 bar, Rh$^0$ is converted into Rh-CO, as suggested by the decrease of the Rh-Rh scattering at 2.5 Å and the increase of the Rh-C scattering at 1.2 Å (Supplementary Table S8, Supplementary Figs. S39 and S40). The corresponding decrease of the normalized XANES intensity at 23221 eV is related to the smaller intensity of this spectral feature in the Rh-CO with respect to the Rh(0) phase. This result is confirmed by STEM and operando IR-MS studies. Indeed, operando IR studies reveal the formation of Rh(I)$(CO)_2$ species under reaction conditions (Supplementary Fig. S42), which is in line with previous studies in the literature where an oxidative disruption of Rh metal clusters in the presence of CO is reported[38]. In addition, both IR-CO (Supplementary Fig. S43) and STEM (Supplementary Fig. S34) show the co-existence of single Rh(I)$(CO)_2$ sites and Rh metal clusters on the sample after having been exposed to reaction conditions, while Rh species in a higher oxidation state, i.e., Rh$^{3+}$ (IR band at 2135 cm$^{-1}$) are not detected in this case. The catalytic performance of the reduced sample in the gas phase hydroformylation of ethylene with syngas at 10 bar (Fig. 4 and Supplementary Figs. S36–S37) displays a

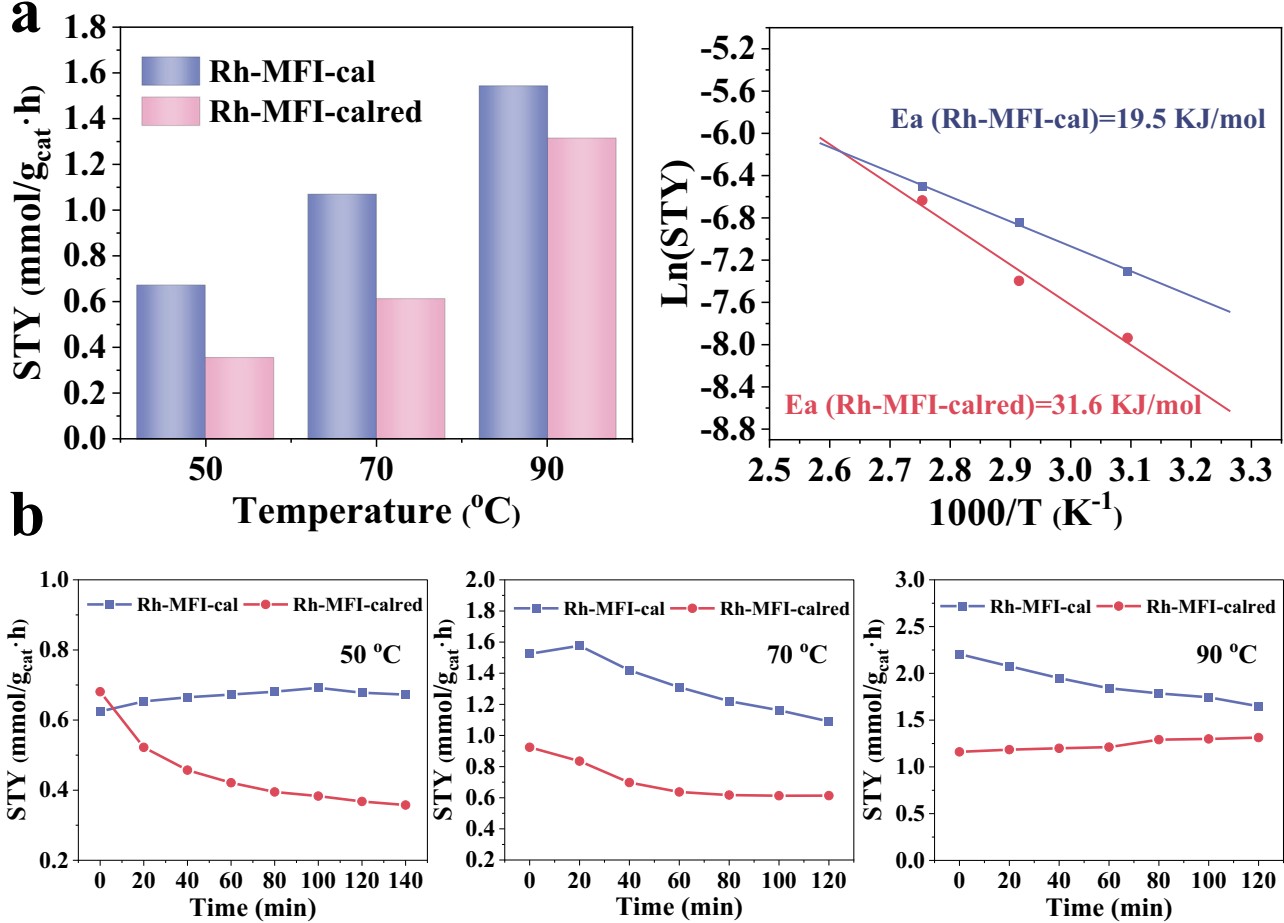

**Fig. 4 | Comparative catalytic and kinetic results of Rh-MFI-cal and Rh-MFI-calred zeolites. a** Catalytic performance of Rh-MFI-cal and Rh-MFI-calred samples in propanal formation under steady state conditions (left) and the respective apparent activation energy (Ea) to propanal formation (right). **b** Propanal space time yield with time on stream at 50 °C (left), 70 °C (middle) and 90 °C (right) on calcined (violet) and calcined-reduced (red) samples. Source data are provided as a Source Data file.

lower reactivity than that of the calcined sample, maintaining 91–96% propanal selectivity in the 50–90 °C range. In particular, the propanal yield is 0.7, 0.9, and 1.3 mmol$_{propanal}$/g$_{cat}$.h at 50, 70 and 90 °C, respectively, and compared to the calcined sample, a higher apparent activation energy for propanal formation (31.6 KJ/mol) is observed, indicating for a less favored process. Spectroscopic studies reveal the in situ formation of Rh(I)(CO)$_2$ species, which, as demonstrated before in the syngas-treated sample, are less active than the previous oxidized species, explaining in this way the catalytic data. In addition, at 50 °C a loss of propanal production with reaction time is observed inferring a lower stabilization of the in situ formed Rh(I)(CO)$_2$ species under reaction conditions, a tendency also observed in the operando XAS studies (Supplementary Fig. S40).

For the sake of comparison, the activity of Rh$^0$ clusters is compared to that of Rh$^0$ nanoparticles (>10 nm). For obtaining Rh nanoparticles, a Rh-impregnated sample on MFI is prepared (details in Supplementary Information 3.4) displaying a homogeneous distribution of Rh nanoparticles around 13 nm (Supplementary Fig. S44). The catalytic performance of this sample is nearly zero, i.e., propanal yields of 0.01, 0.08, and 0.1 mmol$_{propanal}$/g$_{cat}$.h at 50, 70 and 90 °C respectively, and in any case much lower than with the previous Rh-MFI samples (Supplementary Figs. S45–S47), reflecting a low intrinsic activity of Rh NP of particle size above 10 nm under our reaction conditions (50–90 °C, 10 bar). Similar behavior is observed in the IR-MS studies (Supplementary Fig. S48), where by IR-CO it is

demonstrated that big NP behaves inert toward oxidative structural disruption (Supplementary Figs. S49 and S50).

With all the previous results, a dynamic structural rearrangement of the Rh-MFI catalyst under reaction conditions is shown, behaving Rh-MFI samples differently under reaction conditions depending on the pre-activation conditions. Two sites have been identified in our work with different propanal formation rates and activation energies: Oxidized isolated Rh species (in a high oxidation state, presumably as Rh$^{3+}$), formed in situ by disruption of Rh$_2$O$_3$ clusters, and isolated Rh(I)(CO)$_2$ species formed due to oxidative disruption of Rh metal clusters (Figs. 5 and 6). The first one behaves as a more efficient precursor than Rh(I)(CO)$_2$ species in propanal formation. In addition to this dynamic catalyst behavior and the stabilization of high oxidation state Rh$^{3+}$ sites, the zeolite plays an important role in the reaction mechanism as extracted from kinetic studies of the reaction orders in CO and H$_2$ for propanal formation and the apparent activation energies of propanal formation. Kinetic studies are done at low temperatures (50–70 °C) and under differential reaction conditions, i.e., in the kinetic regime (details in Supplementary Information section 4). As mentioned previously, the propanal apparent activation energies for the Rh-MFI samples are in all cases lower than those generally reported in the literature for impregnated Rh-based catalysts[42] (see Supplementary Table S10). This can be explained by the confinement interaction between the metal site and intermediate species and the zeolite framework[47], influencing the adsorption

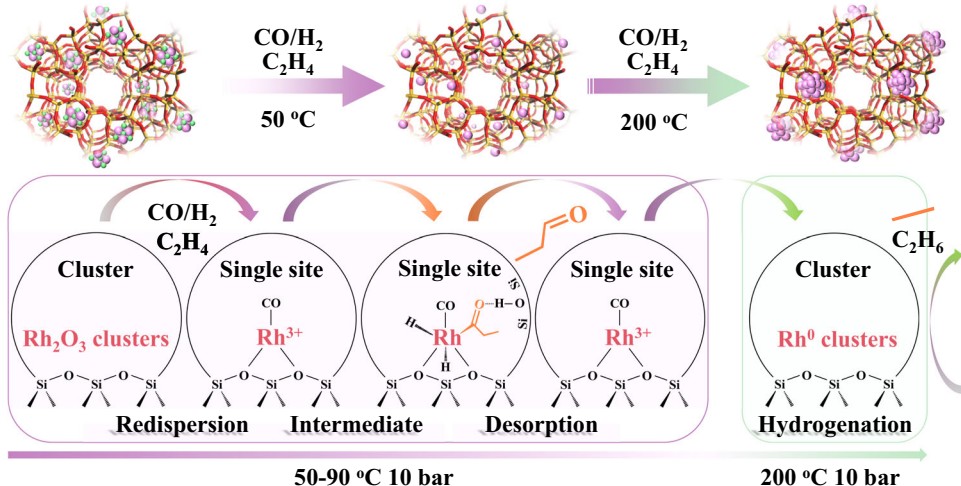

**Fig. 5 | Dynamic behavior of Rh-MFI-cal sample under reaction conditions.** Representative schema showing the disruption of Rh₂O₃ clusters into single Rh³⁺ sites and its stabilization under reaction conditions as Rh(CO)L.

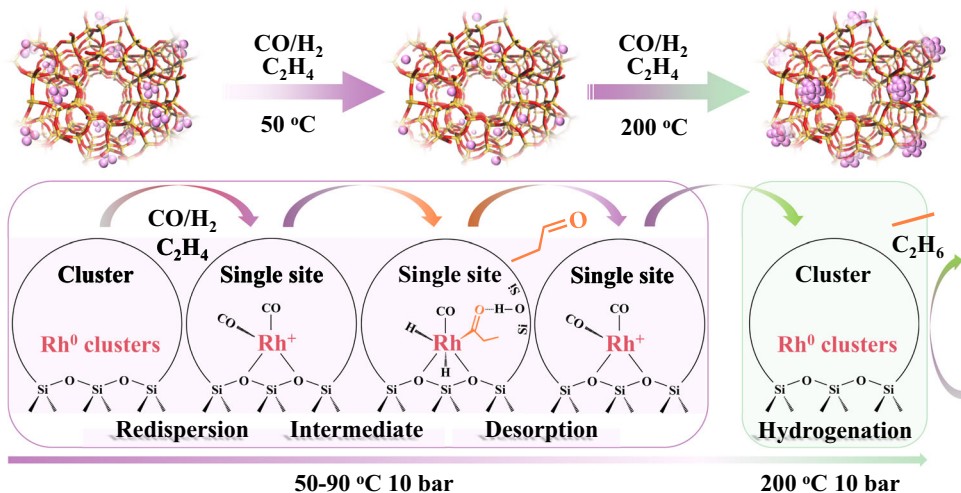

**Fig. 6 | Dynamic behavior of Rh-MFI-calred sample under reaction conditions.** Representative schema showing the disruption of Rh⁰ clusters into Rh(I)(CO)₂ species under reaction conditions.

enthalpies of intermediate products (more discussion in Supplementary Information of section 4). Supporting this idea, the calculated reaction orders in CO and $H_2$ for propanal formation differ between the Rh-MFI samples and a reference Rh/SiO₂-impregnated sample. In the case of Rh-MFI catalysts a positive effect of CO partial pressure on propanal formation is observed independent of the catalyst pre-treatment (see Supplementary Fig. S66), while it is practically zero or slightly negative in the Rh/SiO₂ sample (more details in Supplementary Information, Supplementary Fig. S68), this last value in line with literature data[42,44,48]. This may indicate different interaction strength of CO with Rh species, which depends not only on the oxidation state but also on the local environment of the metal sites. On the other hand, the $H_2$ order is slightly different among the catalysts, from close to 1.5 in the Rh-MFI-cal to 1 in the Rh-MFI-calred and Rh/SiO₂ samples, indicating differences in the H* coordination involvement in the rate-determining step for propanal formation (more discussion in Supplementary Information section 4).

Next, while single Rh sites are stabilized at low reaction temperatures, they tend to aggregate at higher reaction temperatures (from 100 up to 200 °C) into Rh clusters (see Supplementary Figs. S4C, S16, S27 and S40). Reductive agglomeration of single Rh sites in

supported catalysts under hydroformylation conditions at temperatures above 150 °C has already been reported by other authors[24], resulting in a progressive loss of propanal selectivity at the expense of ethane formation by ethylene hydrogenation. A similar behavior is observed in our samples, as displayed in Supplementary Fig. S51 (see also Supplementary Figs. S6 and S37) but, remarkably, the calculated deactivation constant in our catalysts is markedly lower than that reported in other studies (see Supplementary Table S2), which could be associated to the protective role of the zeolite. By correlating spectroscopic and catalytic data at increasing reaction temperatures, metal clusters can be proposed as responsible for ethylene hydrogenation while single sites are responsible for propanal formation, as supported by other works[27]. The calculated apparent activation energy in the 120–200 °C temperature range for the hydrogenation reaction (89.2–131.4 KJ/mol) is higher than that for the hydroformylation reaction (19.5–31.6 KJ/mol)[28,42,49], in line with literature data and with the increase of ethane formation with increasing temperature. Furthermore and in order to determine the extent of irreversible loss of active sites due to metal agglomeration, the catalytic performance (i.e., propanal formation) is analyzed upon lowering back the temperature to 90 °C, after a reaction temperature of 200 °C for 6 h. By doing so, a

35% and 37% loss of activity is observed on the Rh-MFI-cal and Rh-MFI-calred samples, respectively.

## Phosphorous stabilized single Rh sites in a one-pot synthesized Rh-(O)-P-MFI zeolite

From all the above, we conclude that if high selectivity to propanal is desired, an important parameter to consider is the stabilization of Rh single sites. Many attempts have been made for the stabilization of metal sites in supported metal systems, for example, adding specific promoters or forming bimetallic alloys[42,44,50]. In our case, the initial hypothesis was that phosphate ions, if well distributed along the zeolite channels, could stabilize Rh cations avoiding sintering. This led us to design Rh-(O)-P samples using a one-pot strategy that allows a good Rh-(O)-P dispersion and interaction (synthesis details in Supplementary Information of Experimental section 3.5). Spectroscopic characterization including $^{31}P$ NMR and XANES study at the P-K edge shows phosphate $P^{5+}$ ions as the most stable phase (details in Supplementary Information, Supplementary Figs. S57 and S58), whereas the introduction of P did not cause marked morphologic nor electronic differences among the samples, at least from STEM (Supplementary Figs. S52–S55), and IR-CO (Supplementary Fig. S59). However, it has a noticeable effect on the dynamic structural catalyst behavior and stabilization of Rh sites under reaction conditions as shown below. The in situ XAS studies done on the Rh-MFI-calred and Rh-(O)-P-MFI-calred samples, are compared in Fig. 7c–f, where the intensity of the normalized XANES at 23221 eV (corresponding to Rh-CO/Rh(0) phase) and 23240 eV (corresponding to $Rh_2O_3$ phase) as a function of time is reported. The corresponding spectra have been depicted in Supplementary Figs. S40 and S64. In both samples the Rh metal cluster present in the calcined-reduced samples in the presence of syngas is partially disrupted to form Rh-CO bonds, but differently to the Rh-MFI sample, in the Rh-(O)-P-MFI catalyst a higher amount of isolated $Rh^{3+}$ sites are formed under working conditions. The $Rh^{3+}$ single site formation is evidenced by an increased intensity of the XANES spectra at 23240 eV (Fig. 7e, f) and the increase of the contribution at 1.5 Å (Rh-O shell) in the FT (Supplementary Fig. S64). The fact that no signal is detected around 2.8 Å in the FT suggests the absence of Rh-Rh $_{Rh2O3}$ clustering (Supplementary Fig. S64). In more detail, IR-CO studies reveal the additional stabilization of oxidized $Rh^{3+}$ sites together with $Rh(I)(CO)_2$ under reaction conditions in the presence of P (Fig. 7b and details in Supplementary Information). Based on our results, it is believed that $Rh^{3+}$ is stabilized via Rh-O = P or Rh-O(H)-P interaction, (see Fig. 8). The promoting effect of high oxidized Rh single site (i.e., $Rh^{3+}$) is confirmed in the catalytic studies, resulting in higher propanal formation. Thus at 90 °C the propanal yield increased from 1.3 up to 1.8 $mmol_{propanal}/g_{cat}$.h in the presence of P (Fig. 7a), behavior also confirmed by operando IR-MS studies (Supplementary Fig. S65). In fact, the stabilization of oxidized $Rh^{3+}$ species in the Rh-(O)-P-MFI-calred sample, decreases slightly the apparent activation energy of propanal formation (Supplementary Table S10). It is then proposed that P behaves as a promoter, stabilizing oxidized (i.e., $Rh^{3+}$) species during the oxidative disruption of Rh clusters which takes place under reaction conditions, and accordingly retarding metal sintering (Fig. 8).

The stabilization effect of oxidized Rh species by P is more evident at 200 °C, conditions where Rh single sites are demonstrated to be unstable and tend to agglomerate with a corresponding loss of propanal activity. In particular, at those conditions, propanal yields of 7.0 $mmol_{propanal}/g_{cat}$.h are obtained in the P-doped Rh-MFI sample, being ~1.3 and ~2.1 times higher than in the un-doped calcined and calcined-reduced samples (Fig. 9). Moreover, the extent of irreversible loss of active sites at the highest operation temperature in the Rh-(O)-P-MFI-calred samples is lower than in the un-doped sample, pointing to a 46% restoring of the catalytic activity upon lowering the reaction temperature back to 90 °C compared to the value of 37% in the un-doped sample.

In contrast to ethylene, propylene is a more challenging substrate, where regio-selectivity to linear butaldehyde remains a great issue[27,51–53]. In this study, the gas phase propylene hydroformylation is studied at 1 bar, GHSV of 10500 $h^{-1}$ and $C_3^=/CO/H_2/N_2$ molar ratio of 0.6/0.6/0.6/1 (details in Supplementary Information of Experimental section 5). As shown in Supplementary Table S14, at 120 °C, the linear to branched molar ratio to aldehydes (l/b) is 4.17 in the Rh-MFI-cal sample with a total selectivity to aldehydes of 67.6%, whereas in the Rh-(O)-P-MFI sample the l/b is 2.42 and the selectivity to aldehydes of 54.3% at the same temperature. Both values are slightly higher compared to state-of-the-art phosphine-free Rh-based catalysts operating under atmospheric gas phase conditions (see Supplementary Table S14). It is proposed that steric effects imposed by the zeolite may restrict the adsorption configuration of propylene enhancing the regio-selectivity to linear aldehyde. This result corroborates the effective interaction between Rh species and the zeolite framework, meanwhile, a clear effect of phosphate ions in promoting regio-selectivity is not observed. In this respect, it has to be said that the linear to branched molar ratio to aldehydes is influenced not only by steric effects but also by the electronic properties of the catalyst.

## Discussion

In summary, by combining multiple operando spectroscopic tools with nanoscale imaging techniques, we show how the morphology of the catalyst can be strongly affected by the reaction environment and how this can affect the catalytic performance. This is very important in the case of structure-sensitive reaction, such as hydroformylation of olefins with syngas, where catalyst restructuration need to be taken into consideration when discussing catalytic results. In particular, an oxidative disruption of zeolite-confined $Rh^0$ clusters into $Rh(I)(CO)_2$ single sites is observed, while it is absent on big Rh particles. A different scenario is observed in the case of zeolite-confined $Rh_2O_3$ clusters which under reaction conditions evolve into highly oxidized single Rh sites, tentatively assigned as $Rh^{3+}$ based on IR-CO of the used catalyst. In both cases, besides the predominance of single sites, the co-existence of ultra-small Rh clusters cannot be discarded. The catalytic performance of the Rh-MFI catalyst in ethylene hydroformylation with syngas is intimately connected with the electronic and oxidation state of Rh sites, demonstrating in this study that oxidized $Rh^{3+}$ site behaves as a more effective precursor for low-temperature propanal formation than commonly reported $Rh(I)(CO)_2$ species. However, a major limitation in all cases is the agglomeration of single sites at high temperatures resulting in Rh clusters which are active sites for ethane formation, and, if sintering is more severe, it results in Rh NP, with a dramatic loss of activity. We have shown by using a one-pot synthesis method employing a P-containing template in the zeolite synthesis, that the formation of phosphate ions within the pores of the zeolite, is an interesting strategy for stabilization of oxidized Rh sites and maintaining catalyst activity and selectivity.

In definitive, besides the intrinsic fundamental interest this work may have in the hydroformylation process promoting the design of high-performance catalysts, it has a wider interest since it shows an effective way to stabilize single Rh sites with different oxidation states inside the channels of the MFI zeolite. We demonstrate the possibility of stabilizing $Rh^{3+}$ with phosphorous that is introduced during the zeolite synthesis in the form of a phosphonium zeolite template. This will allow the synthesis on demand of efficient catalysts for specific reactions.

## Methods
### Chemical reagents
All chemicals and materials were obtained from commercial sources and used without further purification unless otherwise specified. Tetrapropylammonium bromide (TPABr), tetraethylphosphonium bromide (TEPBr), tetraorthoethylsilicate (TEOS), ethylenediamine,

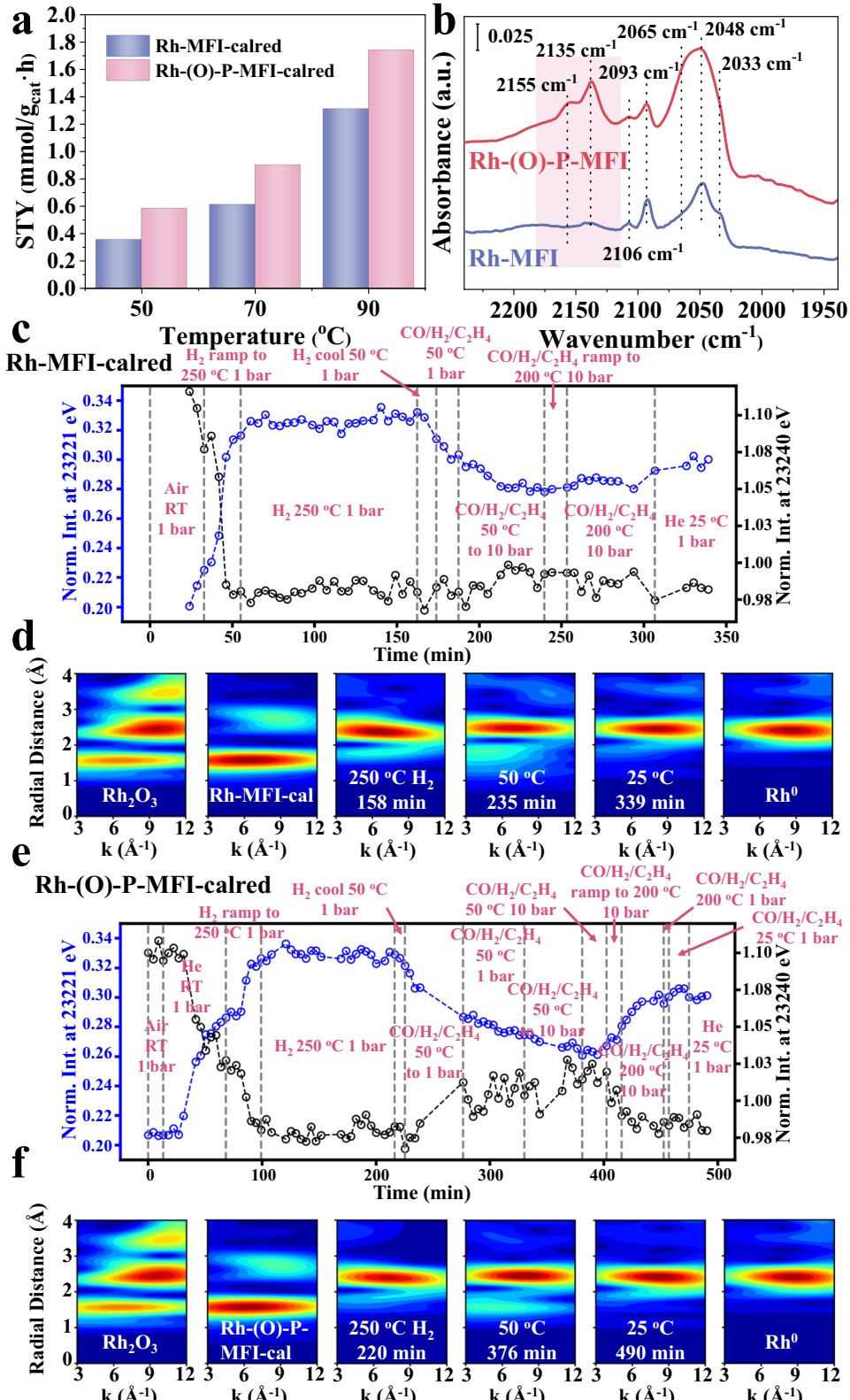

**Fig. 7 | Catalytic performance and spectroscopic characterization of Rh-(O)-P-MFI zeolite. a** Promoting effect of P in the catalytic performance. **b** IR-CO at −65 °C of samples after exposure to hydroformylation, Rh-(O)-P-MFI-calred (red line), Rh-MFI-calred (violet line), showing stabilization of Rh³⁺ in the presence of P. **c**−**f** Evolution of whiteline intensities at 23221 eV corresponding to Rh-CO/Rh(0)

(blue) and 23240 eV corresponding to Rh₂O₃ (black) for Rh-MFI-calred and Rh-(O)-P-MFI-calred, as well as Cauchy wavelet, transforms at key time points upon exposure to reaction conditions after H₂ treatment. Source data are provided as a Source Data file.

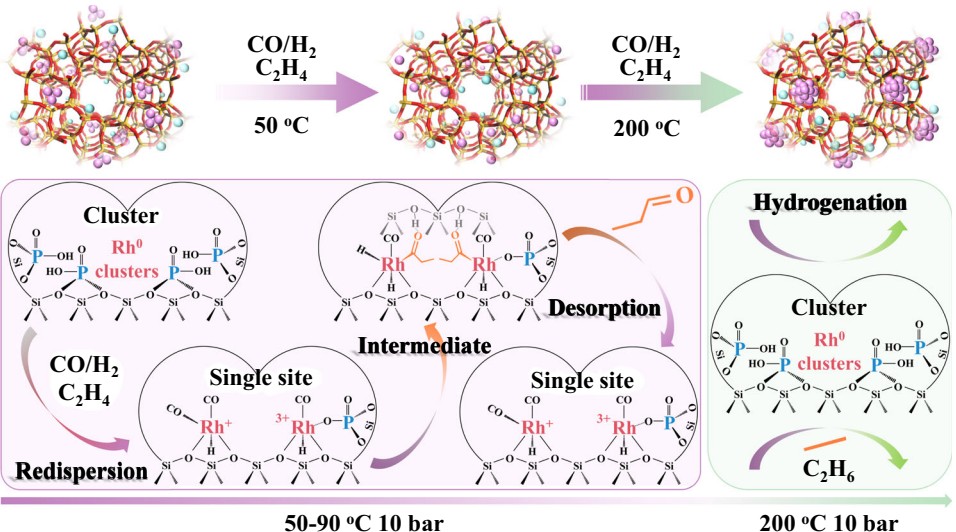

**Fig. 8 | Dynamic behavior of Rh-(O)-P-MFI-calred sample under reaction conditions.** Representative schema showing the disruption of Rh⁰ clusters into single Rh sites with the stabilization of Rh³⁺ sites by adjacent phosphate ions.

rhodium (III) chloride hydrate, $SiO_2$ (Aerosil 200) and anion exchange resin were purchased from Sigma-Adrich, 99.9% purity. Tetrapropylammonium hydroxide (TPAOH) and tetraethylphosphonium hydroxide (TEPOH) were obtained by ionic exchange method with anion exchange resin. Deionized (DI) water from Milli-Q integral water purification system (Millipore, 18.2 MΩ·cm⁻¹).

## Catalyst synthesis

Rh-MFI is synthesized by conventional hydrothermal synthesis with tetraorthoethylsilicate (TEOS) as a silica source and tetrapropylammonium hydroxide (TPAOH) as an organic template. The rhodium precursor is a solution of rhodium (III) chloride hydrate and ethylenediamine (Rh-en). In a typical synthesis, the solution of rhodium precursor is prepared by dissolving 0.263 g RhCl₃ hydrate (38% Rh) in 90 g milliQ water. When the solution became homogeneous, 10 g ethylenediamine was added to the solution under stirring. The solution is kept stirring at room temperature for 16 h until a transparent light yellow solution is formed. Then, 9.5 g TPAOH (43.12%, exchanged from tetrapropylammonium bromide in-house using anion exchange resin) and 10 g H₂O are mixed in a plastic beaker, followed by adding of 17.36 g TEOS. Then, the beaker is sealed with parafilm and kept under stir for 2 h to allow full hydrolysis of TEOS. Afterward, the parafilm is removed and the solution is allowed to keep under stir for 16 h to evaporate ethanol generated from the hydrolysis of TEOS. Then, 5 g Rh-en solution is added to the zeolite synthesis gel and is transferred to a Teflon-lined autoclave and heated to 175 °C under agitation for 4 days for crystallization. When the crystallization is finished, the solid product is collected by filtration and washed thoroughly with distilled water and dried in air at 100 °C. The dried sample is calcined in a tubular oven in air (75 mL/min) with a heating rate of 1.5 °C/min to 550 °C and maintained 6 h. The obtained sample is named Rh-MFI-cal.

The calcined Rh-MFI sample is reduced in H₂ with a heating rate of 10 °C/min to 600 °C and maintained for 3 h. The obtained sample is named Rh-MFI-calred.

Synthesis of Rh-(O)-P-MFI is in a similar manner as the Rh-MFI sample, with tetraorthoethylsilicate (TEOS) as silica source and a mixture of tetrapropylammonium hydroxide (TPAOH) and tetraethylphosphonium hydroxide (TEPOH) as organic templates. The rhodium precursor is a solution of rhodium (III) chloride hydrate and ethylenediamine (Rh-en). In a typical synthesis, the solution of rhodium precursor is prepared by dissolving 0.263 g RhCl₃ hydrate (38%

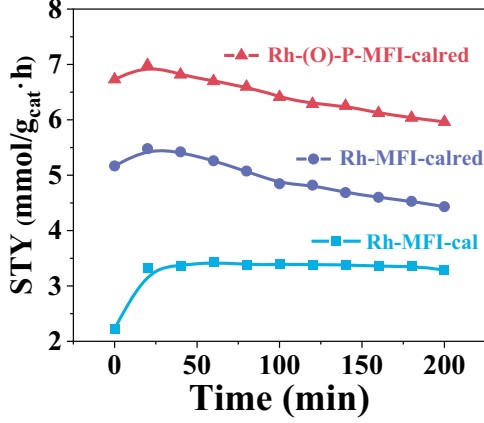

**Fig. 9 | Comparative catalytic performance of Rh-MFI-cal, Rh-MFI-calred and Rh-(O)-P-MFI-calred zeolites at 200 °C.** Catalytic performance at 200 °C and 10 bar on the Rh-MFI-cal (blue), Rh-MFI-calred (violet) and Rh-(O)-P-MFI-calred (red) samples. Source data are provided as a Source Data file.

Rh) in 90 g milliQ water, when the solution becomes homogeneous, 10 g ethylenediamine is added into the solution under stirring. The solution is kept stirring at room temperature for 16 h till a transparent light yellow solution is formed. Then, a mixture containing 9.5 g TPAOH (43.12%, exchanged from tetrapropylammonium bromide in-house using anion exchange resin), 1.75 g TEPOH (9.37%) and 10 g H₂O is mixed in a plastic beaker, followed by adding of 17.36 g TEOS. Then, the beaker is sealed with parafilm and kept under stir for 2 h to allow full hydrolysis of TEOS. Afterward, the parafilm is removed and the solution is allowed to keep under stir for 16 h to evaporate ethanol generated from the hydrolysis of TEOS. Then, 5 g Rh-en solution is added to the zeolite synthesis gel and is transferred to a Teflon-lined autoclave and heated to 175 °C under agitation for 4 days for crystallization. When the crystallization is finished, the solid product is collected by filtration and washed thoroughly with distilled water and dried in air at 100 °C. The obtained sample is named Rh-(O)-P-MFI-cal. The dried sample is calcined in a tubular oven in air (75 mL/min) with a heating rate of 1.5 °C/min to 550 °C and maintained for 6 h. Then, it is reduced in H₂ with a heating rate of 10 °C/min to 600 °C and maintained for 3 h. The obtained sample is named Rh-(O)-P-MFI-calred.

For reference catalyst, Rh is introduced by the impregnation method. First, pure-Si MFI zeolite is synthesized using the same procedure described for the one-pot sample, only excluding the addition of the Rh precursor in the solution. When the synthesis is finished, the sample is calcined in a tubular oven in air (75 mL/min) with a heating rate of 1.5 °C/min to 550 °C and maintained for 6 h. Then a solution of RhCl₃ containing the required amount of Rh for a final loading of 0.3 wt% and incipient wetness amount of water is added and mixed with the solid and dried in air at 100 °C to remove the physically adsorbed water. The dried sample is calcined again in a tubular oven in air (75 mL/min) with a heating rate of 10 °C/min to 550 °C and maintained for 3 h. The calcined sample is named Rh/MFI-cal. Part of the Rh/MFI-cal sample is reduced in H₂ with a heating rate of 1.5 °C/min to 600 °C and maintained for 3 h. This prepared sample is named Rh/MFI-calred.

Next, 0.1 wt% Rh impregnated on SiO₂ is prepared by the same method to impregnated Rh/MFI, except the support is amorphous Aerosil 200.

## Characterizations

Samples for electron microscopy studies were prepared by dropping the suspension of Rh-MFI or Rh-(O)-P-MFI catalysts, using $CH_2Cl_2$ as the solvent, directly onto holey-carbon-coated Cu grids. The measurements were performed using a JEOL 2100 F microscope operating at 200 kV, both in transmission (TEM) and scanning-transmission modes (STEM). HR-HAADF-STEM and iDPC imaging were performed on a double aberration-corrected, monochromated FEI Titan3 Themis 60-300 microscope operating at 300 kV. All calcined-reduced samples were quickly removed to glove box after reduction and the loading of catalysts on the Cu grid for TEM measurements were also operated in glove box to avoid air oxidation. Inductively Coupled Plasma Mass Spectrometry (ICP) was carried out with a Varian 715-ES ICP-Optical Emission spectrometer and analyzed with a SCHN FISONS elemental analyzer. Textural properties like BET surface area, micropore volume and external surface area were measured on Micromeritics ASAP2000. Solid-state ³¹P MAS NMR spectra were recorded at room temperature under magic angle spinning in a Brucker AV-400 spectrometer. IR spectra of CO adsorbed on Rh-MFI and Rh-(O)-P-MFI catalysts were recorded with Bruker 70 V spectrometer under −65 °C. Operando IR-MS were performed in a commercial IR catalytic cell (Aabspec) connected "online" to a mass spectrometer (MS) (Balzer (QMG 220 M1)) under 9 bar of reaction gas. In situ XAS spectroscopy was acquired at CLÆSS beamline of the ALBA synchrotron. The measurement of X-ray absorption near-edge structure (XANES) of Rh-(O)-P-MFI catalyst was performed at beamline 9-BM of the Advanced Photon Source (APS) (Argonne National Laboratory, Argonne, Illinois). More details can be seen in Supplementary Information.

## Catalytic activity evaluation

The catalytic activity for ethylene hydroformylation was evaluated in a fixed-bed reactor with an inner diameter of 6.5 mm and length of 133 mm, in the temperature range of 50–200 °C at 10 bar. A synthetic syngas mixture CO/H₂/Ar (45/45/10, vol%), ultra-high purity-grade C₂H₄ and N₂ were mixed in a molar ratio 1:1:1:0.5 (CO/H₂/C₂H₄/N₂) to reach a total flow rate 35 mL/min for reaction. The line associated with the CO/H₂/Ar mixture was equipped with a carbonyl trap containing activated carbon upstream of the mass flow controller to retain the metallic carbonyls that might be formed in pressurized gas bottles. Before reaction, Rh-MFI-cal was freshly calcined and directly put for reaction. Syngas pre-activated Rh-MFI-cal was in situ activated at 120 °C for 2 h under 22 mL/min CO/H₂/Ar (45/45/10, vol%) at atmospheric pressure with a heating rate of 10 °C/min. Rh-MFI-calred and Rh-(O)-P-MFI-calred were in situ reduced at 250 °C for 2 h under 20 mL/min H₂ flow at atmospheric pressure with a heating rate of 10 °C/min. Typically, 100 mg catalysts (sieved into 200–400 μm) were diluted with 300 mg SiC granules (Fisher Scientific, 600–800 μm) to achieve an

isothermal packed bed with a gas hour space velocity (GSHV) of 8000 h⁻¹. Under these conditions, mass transfer limitations can be excluded as shown in Supplementary Fig. S1. Moreover, additional experiments were done by modifying the GHSV (800–8000 h⁻¹) in order to increase the ethylene conversion in the 50–120 °C temperature range. The products at the outlet of the reactor were online analyzed with Agilent 8860 equipped with a TCD (HP-Plot/Q plus HP-Molesieve) and an FID (HP-Plot/U) detector. All product lines were heated at 150 °C to prevent condensation of products in the lines. Product quantification was performed using chromatographic response factors referenced to N₂ as an internal standard. Ethylene conversion, product selectivity, space time yield (STY), turnover frequencies (TOF), apparent activation energies (Ea) and carbon balance were calculated according to Eqs. (1)–(6). The elemental balance of carbon was 100 ± 5%. The TOF was calculated using the number of Rh sites assuming that all Rh was atomically dispersed.

$$Conversion \left( X_{C_2H_4} \right) = \left( \frac{F_{C_2H_6\,out} + F_{C_3H_6O\,out} + F_{C_3H_7OH\,out}}{F_{C_2H_4\,out} + F_{C_2H_6\,out} + F_{C_3H_6O\,out} + F_{C_3H_7OH\,out}} \right) * 100(\%) \tag{1}$$

$$S_i = \left( \frac{F_{i\,out}}{F_{C_2H_6\,out} + F_{C_3H_6O\,out} + F_{C_3H_7OH\,out}} \right) * 100(\%) \tag{2}$$

$$STY_i \left( mol \cdot g^{-1} \cdot h^{-1} \right) = \frac{V_{C_2H_4} * X_{C_2H_4} * S_i * 60}{V_m * m_{cat}} \tag{3}$$

$$TOF = \frac{STY_i * M_{Rh}}{m_{cat} * D_{Rh} * W_{Rh}} \tag{4}$$

$$Ln(STY_i) = -\frac{E_a}{RT} + C \tag{5}$$

$$Carbon\ balance = \frac{Total\ carbon\ in\ products}{Total\ carbon\ in\ feed\ components} \tag{6}$$

Where $F_i$ stands for the molar flow of $i$-compound based on the N₂ internal standard using calibrated response factors. $V_{C2H4}$ is the flow of C₂H₄, $V_m$ is the molar volume of an ideal gas at standard temperature and pressure, $M_{Rh}$ is the molar mass of Rh, $D_{Rh}$ is the dispersion of Rh, herein is 1, $W_{Rh}$ is the loading weight of Rh, $R$ is the molar gas constant.

The reaction orders in ethylene hydroformylation were analyzed using a similar procedure as indicated above but operating at low conversion, i.e., high GHSV (18000 h⁻¹). These studies have been done on both Rh-MFI-cal, Rh-MFI-calred, Rh/SiO₂-cal and Rh/SiO₂-calred samples. The reaction orders have been studied at low temperature and under stable conditions (at 50 °C and 70 °C respectively) by varying the partial pressure of one reactant from 0.625 to 3.75 bar while keeping the partial pressure of the other reactant at 2.5 bar.

For propylene hydroformylation reactions, the same fixed-bed setup of ethylene hydroformylation was used. Normally, 30 mg catalysts (sieved into 200–400 μm) were diluted with 300 mg SiC granules (Fisher Scientific, 600–800 μm) to achieve an isothermal packed bed with a GHSV of 10500 h⁻¹. Experiments were done at 1 bar, in the temperature range 50–120 °C and using a total flow of 14 mL/min with gas feed of C₃H₈/CO/H₂/N₂ (0.6/0.6/0.6/1 molar ratio). Equations (1)–(4) were used for the calculation of propylene conversion, product selectivity, and turnover frequencies (TOF).

## Data availability

All data generated in this study are provided in the Supplementary Information and the most relevant one is included in the Source Data file. All relevant data are available from the authors upon request. Source data are provided with this paper.

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

## Acknowledgements
P.C. thanks the Ministerio de Ciencia, Innovación y Universidades, grant number PID2021-1262350B-C31, and Generalitat Valenciana (GVA), grant number CIAICO/2021/2138. P.C. and D.G. thank the Advanced Materials Programme supported by MCIN with funding from European Union Next Generation EU (PRTR-C17.11) (TED2021-130756B-C32) and by Generalitad Valenciana (ref MFA/2022/016). M.Z. acknowledges the China Scholarship Council (CSC) for Ph.D. fellowship (CSC NO. 202006440003). XAS experiments were performed at the ALBA Synchrotron (BL-22 CLÆSS) with the collaboration of ALBA staff. XANES of Rh-(O)-P-MFI samples were analyzed at the Advanced Photon Source, an Office of Science User Facility operated by Argonne National Laboratory for the U.S. Department of Energy (DOE) Office of Science. The U.S. DOE supported it under Contract No. DE-AC02-06CH11357, and the Canadian Light Source and its funding partners. HR-HAADF-STEM measurements were performed at DME-UCA at Cadiz University. M.L.H. and J.J.C. thank the financial support from the Department of Economy, Knowledge, Business and the University of the Regional Government of Andalusia, Project reference FEDER-UCA18-107139.

## Author contributions
A.C., P.C. and M.Z. designed the studies and wrote the paper. M.Z. and C.L. synthesized the catalysts. M.Z. and D.G. did the catalytic test. D.M.M., F.G., V.M.D. and L.S. did the XAS experiments and interpretation. M.L.H. and J.J.C. did the STEM studies and analysis. P.C. did the operando IR study. All authors discussed the results.

## Competing interests
The authors declare no competing interests.
