## [Peer Review File · Nature Communications]

REVIEWER COMMENTS

Reviewer #1 (Remarks to the Author):

This is an interesting paper on the stabilization of Rh(III) sites in MFI involving tetraethylphosphonium hydroxide as template during a one-pot synthesis method. The conclusions seem to be supported by the data and the proposed hydroformylation mechanism seems reasonable. However, I believe that what the manuscript is lacking is computational confirmation of the relative stability of the suggested species as well as of the reaction mechanism. I realise that this might be asking too much of the authors, who have presented a complete work, so I won't hold the manuscript hostage to this. I recommend publication without revisions.

Reviewer #2 (Remarks to the Author):

This manuscript by Zhao et al., entitled "Low temperature hydroformylation of ethylene by single Rh sites stabilized by phosphorous in a one pot synthesis of Rh-P-MFI zeolite", reports the one-pot synthesis of zeolite-immobilized Rh catalyst for gas-phase ethylene hydroformylation. Furthermore, the use of phosphate as precursor of P-based stabilizing agent for rhodium catalyst is demonstrated.

The manuscript is well written and well documented.

A robust catalysis study was performed to understand the evolution of the Rh sites under catalytic conditions.

However, the challenge of ethylene hydroformylation by molecular (heterogenized) rhodium catalyst seems not relevant. Indeed, ethylene as substrate shows no regioselectivity issue in the formation of the corresponding aldehyde and rhodium is known for being highly selective for higher olefins hydroformylation. Propene as more challenging substrate should be included to make this study more appealing and to establish structure-activity relationship in the context of catalytic site evolution as claimed by the authors. Indeed propene will allow mechanistic study based on regioselectivity, not possible with ethene.

In the introduction (line 93-94), the authors claim that their catalyst is "...more active than state-of-the-art catalysts when operating at low temperature (50-100°C), with TOF of 99h⁻¹...". Even if the authors mention a temperature range with 100°C as upper limit, it is noticeable that porous-polymer-based Rh catalyst containing phosphine ligands as been reported with TOF of 10'000h⁻¹ at 120°C, with long term stability and 95+% selectivity for the aldehyde (Jiang et al., J. Mol. Catal. A, 2015, 404-405, 211-217). Introduction has to be updated accordingly and Table S1 has to be updated. Same comment for same claim on line 176-177.

According to characterizations reported line 137 and 159, catalyst is made of both single site mononuclear Rh sites and Rh clusters. How can the authors disentangle the contribution of each species in catalysis?

Also, the crucial stabilizing role of phosphine capping agent in Rh-catalyzed hydroformylation has been widely discussed, including for Rh particles/clusters (see Garcia et al., Appl. Catal. A. Gen, 2017, 548, 136-142). Can the authors comment on their system in light of previous studies?

During catalysis at 70 and 90°C, the Figure 3a shows slow catalyst deactivation. As far as the main asset of this manuscript is the discussion around catalyst evolution, how does the catalyst evolve during reaction under these conditions?

Is ethane, and propanol, production resulting from reaction at rhodium clusters? Or competitive hydrogenation at mononuclear Rh single sites?

Can the authors comment on the effect of CO and H₂ partial pressure? What is the effect of a change in CO:H₂ ratio?

Considering a molecular mechanism at solid's interface, can the authors comment on the role of the support as well as on the dissociation mechanism (energy?) of the Rh₂O₃ to Rh(3+) under reductive atmosphere?

The scheme 1 and 2 are unfortunately not readable and more detailed mechanism for the first step should be postulated.

Finally, in the case of the P-doped catalyst, what is the nature of the Rh-P linkage (and Rh:P ratio in single site) in the "stabilized" catalyst?

A comparison of the authors reported strategy and the encapsulation of commercially available Rh(H)(CO)(PPH₃)₃ should be enlightening.

In conclusion, the novelty of this study lies in the understanding of catalyst nature and evolution under reaction conditions which however needs to be more detailed. Also, the use of heterogenized single-site Rh catalyst for ethylene hydroformylation has been reported several times earlier, with also much higher activity and ethylene is not a challenging substrate for this reaction.

Accordingly, due to questionable novelty, I cannot recommend the manuscript to be published in Nature Communications in its present form. Major revisions addressing at least the comments above are required.

Reviewer #3 (Remarks to the Author):

This work conducted by Zhao et al. involves the synthesis of Rh-MFI catalysts using a one-pot method. These catalysts exhibit high selectivity for the industrially-important hydroformylation reaction at low temperatures (50°C). The addition of phosphate species as a modification shows promise in enhancing the catalyst's stability. Characterization of the catalysts was performed using XAS, IR, and STEM techniques. However, several important questions need to be addressed before considering acceptance in Nat. Commun. This work might be better suited for a specialized catalysis journal.

1. The key novelty lies in the high selectivity of propanal at low temperatures due to the presence of Rh³⁺ species. However, the conversion remains very low (e.g., <1% at 50°C), and the Rh³⁺ species are prone to clustering despite the introduction of P species. At 200°C, the selectivity of hydroformylation drops to approximately 20%, which is not notably impressive compared to previously reported values. Additionally, it is not appropriate to compare the selectivity (Fig 3b), with values reported in the literature when there are significant differences in conversion. Conversion was defined in Methods but it was not reported in this work.

2. The apparent activation barriers for hydroformylation appear to be very low (e.g., 19 kJ/mol). Although these barriers were measured at low conversion with minimal transport limitations, the catalyst deactivation observed at higher temperatures would reduce the measured barriers. Thus, the intrinsic barriers of hydroformylation here should be higher than the measured values, which raises questions when using them for comparison with reported data.

3. The assignment of peaks corresponding to Rh-C and Rh-O bonds in EXAFS is questionable. The indicated positions of these two bonds suggest a bond length difference of ~0.5 Å, which is unlikely in the co-presence of Rh-C bonds from CO adsorption and Rh-O bonds in RhO_x moieties (or with the support). Instead, the apparent two peaks between 1-2 Å (e.g., Fig. S9) are typically the result of strong destructive interference between Rh-C and Rh-O bonds with similar distances and scattering characteristics. The EXAFS fitting should be added and the co-linear scattering of Rh-C=O at a higher shell should also be included, and EXAFS plots in both k and R spaces should be presented. Additionally, it is recommended to perform wavelet-transformed EXAFS analysis to differentiate the presence of Rh-Rh bonds at positions similar to those of the Rh-C=O scattering pathway.

4. The authors deduce small particle size based on peak intensities in FT-EXAFS, yet, which may not necessarily hold true, as peak intensity can also be affected by structural disordering in cluster morphologies. Furthermore, the peak intensity between 1-2 Å is lower than that of Rh₂O₃. Does this imply that Rh^{δ+} is under-coordinated? EXAFS fitting should be performed for the fresh, activated, and reaction conditions.

5. The authors state that Rh is in a high oxidation state based on its similar edge position to that of Rh₂O₃. However, it is not necessarily true, as Rh(I) complex species can also exhibit a similar edge

position to Rh₂O₃. For a better comparison, the XANES spectrum of Rh(I) complex standards, such as those with a local group of Rh(I)(CO)_xO₂, should be included. Moreover, the presence of Rh-O bonds in EXAFS is not exclusively indicative of the presence of Rh³⁺ since isolated Rh⁺ species on the support (e.g., Rh(I)(CO)₂O₂ with a square-planar geometry) can also exhibit Rh-O bonds.

6. In the schema, it was proposed that the monocarbonyl Rh(CO)L complex interacts with OH groups of the zeolite. The OH groups seem to be restored after depressurization and purging with Ar (Fig. S12b). Can it be due to the typical trace amount of water in Argon rather than the dis-interaction with Rh(CO)L?

7. The function by which P stabilizes the Rh sites, whether through electronic, geometric, or steric effects, remains unclear.

8. Abbreviations such as XAS, HAADF, STEM, IR, etc., should be defined upon their first mention. Grammar issues should be addressed approximately. For the sake of readiness, the colors for different samples should be kept consistent such as avoiding the reversal seen in Figs. 4a and 4b.

Reviewer #4 (Remarks to the Author):

In this work, Concepcion, Corma and co-workers report the preparation and mechanistic investigations of highly efficient Rh-MFI catalyst for hydroformylation. The authors also show a one-pot strategy for the preparation of Rh-P-MFI catalyst, in which the oxidized Rh can be stabilized by phosphate ions within the zeolite pores. Spectroscopic tools used in this study further provide insights into the dynamic behaviors of the catalysts. Overall, the new catalysts, preparation strategy and mechanistic investigation in this work present significant findings for the field of catalysis. However, there are some major issues that need to be addressed before the work can be published in Nature Communication:

1. The dynamic behavior proposed in Scheme 1 (and Scheme 3) needs further discussion. The mechanism in which the reaction initiated with a Rh³⁺ species (the second species) seems not follow the classic Heck and Breslow mechanism. Further investigations such as the detection of CO₂ generation at the beginning of hydroformylation and the effect of CO partial pressure on the catalyst activity, could give more convincing conclusions.

2. The apparent activation energies are quite low (19.5 kJ/mol for Rh-MFI-cal). Is it possible that the reaction proceeded with diffusion-controlled kinetics?

3. It is confusing to me that the E_a for ethane formation is comparable and even smaller than propanal formation (Figure S7) at 50-90 °C, whereas the selectivity to aldehyde is >95%. In addition, the E_a values

of ethane measured at 423-473 K are reported in Table S1 and the main article. The authors should comment on their choice of E_a values.

4. The authors should provide more procedure details of the catalyst test. For example, which time period is used for the calculation of STY?

There are also other minor points:

1. In Figure 5d, Rh-MFI-cal is less reactive than Rh-MFI-calred, whereas in the main article Rh-MFI-cal is more active.

2. It is recommended to cite some representative review articles for hydroformylation: e.g., Chem Rev 2012, 5675; Green Synth. Catal. 2021, 247

3. The author should further check their references. For example, ref 34 and 36 are the same.

Reviewer #5 (Remarks to the Author):

In this article, Zhao et al. reported low temperature hydroformylation of ethylene heterogeneously catalyzed by Rh single sites stabilized by phosphorous which is confined inside MFI type zeolite. The high activity of the Rh catalyst is ascribed to the disruption of Rh₂O₃ clusters to the Rh single site. Interestingly, the Rh₀ clusters do not form those active species and hence less active. The authors characterized the catalysts with very high level sophisticated techniques and conducted operando studies to prove whether the true active site is single site or not. There has been quite some discussion on this topic quite recently (doi.org/10.1021/acs.chemrev.2c00495) whether single site remain as it is or it forms something else due to high surface energy and XAS seems to be a valuable technique to track the dynamic evolution of the active site. Discussing such literature within this manuscript will also send a general message to the community that in-situ spectroscopy is highly important for further development of this field. Nevertheless, XAS also has limitations (doi.org/10.1021/acscatal.3c01116), which should also be discussed and hence complementary techniques such as FTIR makes further progress in the field. Hence, this work makes these points clearer.

Overall, the authors did a very elaborative study from catalyst synthesis, reaction kinetics, operando investigation and supported their hypothesis with strong evidences. The study is well presented with a huge amount of data collected. I however have few points of concern which might helps in improving the manuscript and have better understanding from a point of view of a common reader and should be carefully reviewed before this manuscript can be accepted in final form

1. It is quite unclear what is the role of phosphorous as it is highly important as most of the industrial catalysts (molecular Rh complex) requires P in the first coordination shell. Since the authors put it in the title as Rh-P-MFI zeolite and consistently showed that the first shell is Rh-C or Rh-O and this is why I am

confused of the need of P and the title can be mis-leading. It will be nice to elaborate and show distinct evidence that there is no P in the first coordination sphere. The presented ^{31}P NMR and the P K-edge XAS only give information about presence of P.

2. Looking at the IR data, there is a consistent CO vibrational frequency in the range 2040-2060 cm^{-1} which I believe is CO adsorbed over Rh cluster. This vibrational frequency is also observed in the IR-CO at -65°C in figure 1e. Therefore it is clever to assume that there is always a fraction of Rh cluster present in the catalyst which I guess the authors also agrees. Hence the authors should strongly state (especially in the conclusion part) that the possibility of ultra small Rh clusters cannot be completely ruled out. Or if the authors disagree with the statement, then should provide strong evidences.

3. In the XAS spectra, is it correct that the scattering around 2.5 Å is assigned to Rh-Rh or Rh-C≡O? I doubt that Rh-C≡O will show such a strong scattering due to much distortion than the Rh-Rh. The authors should validate this claim by measuring Rh carbonyl complexes with XAS.

4. Even though the authors relate the CO vibrational frequency at 2156 cm^{-1} to the Rh $^{3+}$, in such high frequency many reports were claiming to have weakly adsorbed CO on the support or gas phase CO. Do the authors conducted any blank experiment to check CO adsorption over the zeolite without any Rh. Especially there is also a shoulder peak between 2150-2200 cm^{-1} which is not explained. Further the ratio of the symmetric to antisymmetric peak at 2112 and 2030 cm^{-1} do not seems to follow a trend. I would assume that this ratio should be constant over the time of CO dosing. In all other spectra reported in the electronic supporting information, there is a distinct peak around 2030 cm^{-1} . Is there any reason for that?

Other comments:

1. In scheme 1, 2 and 3, some of the features are very difficult to read. I suggest either less information or bigger font size. For example the way it is shown in page 20 of supplementary information is much clearer. The same goes to figure 1 (c) and (d). X-axis values need larger font size.

2. In page 9, line 175: worth noting instead of worthnothing

3. It is not clear what the authors want to describe in the sentence 254 (page 9) "tendency also observed in the operando XAS studies". What tendency they refer to?

4. Page 17 (line 254) of supporting information, k range is represented as Å $^{-1}$ instead of Å.

The authors appreciate the reviewer's effort and time for the critical review of the article. The manuscript has been update with new information and a point to point answer to all the comments risen by the reviewers is included in this document.

All changes done in the manuscript are marked in yellow in the revised version.

Reviewer #1 (Remarks to the Author):

Comments:

This is an interesting paper on the stabilization of Rh(III) sites in MFI involving tetraethylphosphonium hydroxide as template during a one-pot synthesis method. The conclusions seem to be supported by the data and the proposed hydroformylation mechanism seems reasonable. However, I believe that what the manuscript is lacking is computational confirmation of the relative stability of the suggested species as well as of the reaction mechanism. I realise that this might be asking too much of the authors, who have presented a complete work, so I won't hold the manuscript hostage to this. I recommend publication without revisions.

Answers: We thanks the referee for his/her positive comments.

Reviewer #2 (Remarks to the Author):

Comments:

This manuscript by Zhao et al., entitled "Low temperature hydroformylation of ethylene by single Rh sites stabilized by phosphorous in a one pot synthesis of Rh-P-MFI zeolite", reports the one-pot synthesis of zeolite-immobilized Rh catalyst for gas-phase ethylene hydroformylation. Furthermore, the use of phosphate as precursor of P-based stabilizing agent for rhodium catalyst is demonstrated. The manuscript is well written and well documented.

A robust catalysis study was performed to understand the evolution of the Rh sites

under catalytic conditions. However, the challenge of ethylene hydroformylation by molecular (heterogenized) rhodium catalyst seems not relevant. Indeed, ethylene as substrate shows no regioselectivity issue in the formation of the corresponding aldehyde and rhodium is known for being highly selective for higher olefins hydroformylation. Propene as more challenging substrate should be included to make this study more appealing and to establish structure-activity relationship in the context of catalytic site evolution as claimed by the authors. Indeed propene will allow mechanistic study based on regioselectivity, not possible with ethene.

Answers: According to the reviewer suggestions, we have done additional experiments using propylene as substrate. The results have been included in the manuscript in Page 19 and in the SI (Section 5).

PAGE 19

In contrast to ethylene, propylene is a more challenge substrate, where regioselectivity to linear-butaldehyde remains a great issue. We have followed the recommendation of the referee, and for the revised version we have performed the gas phase propylene hydroformylation at 1 bar, GHSV of 10500 h⁻¹ and C₃=/CO/H₂/N₂ molar ratio of 0.6/0.6/0.6/1 (details in SI). As shown in Table S14, at 120 °C, the linear to branched molar ratio to aldehydes (l/b) is 4.17 in the Rh-MFI-cal sample with a total selectivity to aldehydes of 67.6%, whereas in the Rh-(O)-P-MFI sample the l/b is 2.42 and the selectivity to aldehydes of 54.3%. Both values are some higher compared to State-of-the-art phosphine free Rh based catalysts operating under atmospheric gas phase conditions (see Table S14). It is proposed that steric effects imposed by the zeolite, may restrict the adsorption configuration of propylene enhancing the regioselectivity to linear aldehyde. This result corroborate the effective interaction between Rh species and the zeolite framework, meanwhile a clear effect of phosphate ions in promoting regioselectivity is not observed. In this respect it has to be said that the linear to branched molar ratio to aldehydes is influenced not only by steric effects but also due to electronic properties of the catalyst.

We appreciate very much the comment of the referee. Nevertheless, we need to consider that the nature of active species and the fragmentation of metal clusters in propylene hydrofomylation cannot be directly extrapolated from that observed in ethylene hydroformylation. Thus, a more detailed study combining operando XAS, IR and HRTEM will be done in the future to understand the influence of steric and electronic properties of Rh species on the regioselectivity to aldehyde. According to the suggestion of the reviewer we have included in this work this

preliminary study, showing promising results that will be completed with more detail and extension in future work.

SI. SECTION 5. Hydroformylation of propylene.

Table S14 shows the catalytic performance of Rh-MFI and Rh-(O)-P-MFI catalysts in the hydroformylation of propylene, where (l/b) represent the linear to branched molar ratio of aldehydes. Comparison with state of the art phosphine free catalysts is also included.

Table S14. Catalytic performance of Rh-MFI-cal and Rh-(O)-P-MFI-calred catalysts in the gas phase hydroformylation of propylene. (l/b) represent the linear to branched molar ratio of aldehydes. State-of-the-art heterogeneous phosphine free catalyst are also included

Catalyst	Contact time (mg·min·ml ⁻¹)	GHSV (h ⁻¹)	T (°C)	P (bar)	Conversion (%)	Selectivity to aldehydes (%)	TOF (h ⁻¹)	l/b	Reference ^d
Rh-MFI-cal	2.14	10500	50	1	0.06	3.95	0.38	2.23	This work
			90	1	0.07	49.45	5.26	3.56	This work
			120	1	0.18	67.64	18.31	4.17	This work
Rh-(O)-P-MFI-calred	2.14	10500	50	1	0.06	11.35	0.76	2.08	This work
			90	1	0.10	43.05	4.65	2.37	This work
			120	1	0.21	54.29	12.18	2.42	This work
Rh/SiO ₂	21.33	-	140	1	0.20	42	0.10	2.33	38
Rh-Li/SiO ₂	21.33	-	140	1	0.85	59	0.64	2.70	38
Rh-K/SiO ₂	21.33	-	140	1	0.55	48	0.34	3.35	38
Rh-Rb/SiO ₂	21.33	-	140	1	0.34	46	0.21	4.00	38
Rh-Mo/SiO ₂	16.67	3600	145	1	0.65	39 ^b	88.1 ^b	-	17
Rh/SiO ₂	16.67	3600	160	1	0.05	40	0.66	6.50	39
Rh-V-SiO ₂	16.67	3600	160	1	0.39	3	0.24	1.5	39
RhCo ₃ /SiO ₂	3	-	160	1	< 2	47 ^c	~1.20	3.13	21

^a All hydroformylation reactions were conducted under gas-solid conditions and $H_2/CO=1:1$. Batch reactor reaction and phosphine-based catalysts were not summarized in this table.

^b Only n-butanol and iso-butanol are the products. Selectivity of aldehydes below 1%.

^c Aldehydes and alcohols are included.

^d References in the new version of the SI

Comments:

In the introduction (line 93-94), the authors claim that their catalyst is "...more active than state-of-the-art catalysts when operating at low temperature (50-100°C), with TOF of 99h⁻¹...". Even if the authors mention a temperature range with 100°C as upper limit, it is noticeable that porous-polymer-based Rh catalyst containing phosphine ligands as been reported with TOF of 10'000h⁻¹ at 120°C, with long term stability and 95+% selectivity for the aldehyde (Jiang et al., J. Mol. Catal. A, 2015, 404-405, 211-217). Introduction has to be updated accordingly and Table S1 has to be updated. Same comment for same claim on line 176-177.

Answers: We thanks the referee for this valuable comment. We have update Table S1, now Table S2 and included a new Table S3, with new references and modify the discussion in the manuscript.

PAGE 5 "The resultant Rh-MFI calcined catalyst is more active compared to state-of-the-art phosphine free solid catalysts when operating at low temperature (50-100 °C), with TOF of 99 h⁻¹ at 90 °C and ~92 % selectivity to the aldehyde".

PAGE 10 "which are sensibly higher than those reported in the literature for phosphine free solid catalysts in the hydroformylation of ethylene (see Table S2 and S3 and Figure 3b)"

PAGE 10 in Figure 3b caption 3D map of TOF and propanal selectivity versus temperature of state-of-the-art-phosphine free solid catalysts (references in Table S2).

Comments:

According to characterizations reported line 137 and 159, catalyst is made of both single site mononuclear Rh sites and Rh clusters. How can the authors disentangle the contribution of each species in catalysis?

Answers: The quantification of the contribution of each specie is challenging, since a mixture of different species are always present, while in different proportion. Thus, at low temperature single Rh sites predominate coming from the fragmentation of Rh₂O₃ or Rh⁰ clusters, while the presence of ultra-small metal clusters cannot be excluded. At increasing temperature, single sites aggregate with the formation of bigger metal clusters. The fact that propanal selectivity is > 95% at low temperature and decays to

almost 20% at increasing the temperature up to 200 °C, while ethane formation increases from < 3% at 50 °C to ≥ 80% at 200 °C, can indicate that single site are responsible for propanal formation whereas metal clusters better catalyse ethane formation. Similar results have also been reported in the literature by other authors (*Angew. Chem. Int. Ed.* **62**, e202214048, 2023; *J. Am. Chem. Soc.*, **145**, 2023,2911-2929; *ACS Catal.*, **9**, 2019, 10899-10912).

Comments:

Also, the crucial stabilizing role of phosphine capping agent in Rh-catalyzed hydroformylation has been widely discussed, including for Rh particles/clusters (see *Garcia et al., Appl. Cata. A. Gen.*, 2017, 548, 136-142). Can the authors comment on their system in light of previous studies?

Answers: As well indicated by the referee, Rh based catalysts incorporating phosphine or phosphite ligands have been widely studied in alkene hydroformylation due to its high activity and selectivity, which is attributed to the existence of a strong phosphine-Rh interaction. We have to indicate that the goal of our work was not to obtain phosphine like complexes, but, instead, generate phosphate ions by a one pot synthesis procedure, as a potential ligand for stabilization of Rh single sites. The formation of phosphate ions has been confirmed by ³¹P NMR where characteristic peaks at -5.8 and -16.2 ppm are detected, and by XANES analysis at the P K-edge showing that P oxidation state is 5+. Under reaction conditions, Rh ions formed via disruption of the Rh cluster interact with nearby phosphate ions, stabilizing via O or O (H) interaction high oxidised Rh species. This is supported by operando XAS and IR studies showing the stabilization of Rh³⁺ species in the phosphorous promoted Rh-MFI sample when exposed to reaction conditions. Based on all these results, we propose a coordination of Rh species with phosphate ions via “O interaction” (i.e., P=O---Rh or P-O(H)---Rh). Phosphate have less negative charge on the oxygen than phosphine oxides (*Eur. J. Inorg. Chem.*, 2019, 1679-1687), and their interaction with Rh is completely different from that of phosphine ligands (through the P atom lone pair of electrons).

Comments:

Figure 3a shows slow catalyst deactivation. As far as the main asset of this manuscript is the discussion around catalyst evolution, how does the catalyst evolve during reaction under these conditions?

Answers: From operando spectroscopic studies we observed a small sintering of Rh single sites with a slight increase of small Rh(0) clusters when operating at 90 °C. This has been observed by STEM (Figure S4) and by EXAFS data (Figure S14), explaining in that way the slow catalyst deactivation of Figure 3a. This has already been commented in the manuscript.

Comments:

Is ethane, and propanol, production resulting from reaction at rhodium clusters? Or competitive hydrogenation at mononuclear Rh single sites?

Answers: That is a good question. According to Marin et al (*Appl. Catal. A; General*, 469, 2014, 357-366) two parallel reactions are proposed, one involve CO insertion leading to aldehyde or alcohol formation, and the other one is ethylene hydrogenation into ethane. But it is not clear on which active site propanol is formed. Based on the operando IR-mass profile presented in this work, propanol grows in parallel to ethane formation once metal clusters predominate on the catalyst. Therefore we believe that metal clusters are involved in the hydrogenation path while single sites better catalyse the hydroformylation path.

Comments:

Can the authors comment on the effect of CO and H₂ partial pressure? What is the effect of a change in CO:H₂ ratio?

Answers: Following the referees suggestions, additional experiments have been done where the effect of CO and H₂ partial pressures are studied. These studies have been done on both Rh-MFI-cal and Rh-MFI-calred samples operating under stable conditions (at 50 °C and 70 °C respectively) and 10 bar by varying the H₂/CO ratio, while the partial pressure of C₂H₄ remained constant. The measured rates were fitted by a power law kinetic model following the expression: $r_{\text{Propanal}} = k \cdot P_{\text{H}_2}^\alpha \cdot P_{\text{CO}}^\beta \cdot P_{\text{C}_2\text{H}_4}^\gamma$, where $P_{\text{C}_2\text{H}_4}^\gamma$ is constant, therefore the expression is rewritten as $r_{\text{Propanal}} = k' \cdot P_{\text{H}_2}^\alpha \cdot P_{\text{CO}}^\beta$, where $k' =$

$k_{\text{PC}_2\text{H}_4}$, α and β correspond to the apparent reaction orders with respect to H_2 and CO partial pressures, respectively. The conversion of ethylene was kept under 1% so that the reaction operates under differential conditions, that is, in the kinetic regime. Thus, it is likely that the apparent orders here are representative of the direct rate of the kinetically relevant step of the reaction mechanism.

Figure S62 shows the effect of the partial pressure of CO and H_2 on the rates of propanal formation on the Rh-MFI-cal (violet line) and Rh-MFI-calred (red line) samples.

Figure S62. Reaction orders for Rh-MFI-cal (violet) and Rh-MFI-calred (red) catalysts at 50 °C and 70 °C, respectively. Rates for hydroformylation to propanal formation as a function of (a) H_2 partial pressure ($P_{\text{CO}} = 2.5$ bar, $P_{\text{C}_2\text{H}_4} = 2.5$ bar) and (b) CO partial pressure ($P_{\text{H}_2} = 2.5$ bar, $P_{\text{C}_2\text{H}_4} = 2.5$ bar) for a total operating pressure of 10 bar.

In addition, the influence of the CO and H_2 partial pressure, and different ratios of $\text{H}_2/\text{CO}/\text{C}_2\text{H}_4$ on the formation rate to propanal is displayed in Table S12.

Table S12. Effect of the partial pressure of H₂ and CO, and the ratio H₂/CO/C₂H₄ on the formation rate of propanal (r, mol_{Propanal}/g_{cat}·h) in the hydroformylation of ethylene on Rh-MFI cal and Rh-MFI calred catalysts. Reaction conditions: 50 °C (Rh-MFI-cal) and 70 °C (Rh-MFI-calred), 10 bar and gas mixture flow 80 mL/min.

Catalyst	H ₂ /CO/C ₂ H ₄	Partial Pressure (bar)					r _{propanal} (mol _{Propanal} /g _{cat} ·h)
		P _{CO}	P _{H2}	P _{C2H4}	P _{Ar}	P _{N2}	
Rh-MFI-cal	1/4/4	0.625	2.500	2.500	3.125	1.250	1.8x10 ⁻⁴
	1/1/1	2.500	2.500	2.500	1.250	1.250	5.5x10 ⁻⁴
	1.5/1/1	3.750	2.500	2.500	0.000	1.250	8.4x10 ⁻⁴
	4/1/4	2.500	0.625	2.500	3.125	1.250	7.8x10 ⁻⁵
	1/1/1	2.500	2.500	2.500	1.250	1.250	5.2x10 ⁻⁴
	1/1.5/1	2.500	3.750	2.500	0.000	1.250	1.1x10 ⁻³
Rh-MFI-calred	1/4/4	0.625	2.500	2.500	3.125	1.250	1.9x10 ⁻⁴
	1/1/1	2.500	2.500	2.500	1.250	1.250	4.6x10 ⁻⁴
	1.5/1/1	3.750	2.500	2.500	0.000	1.250	8.1x10 ⁻⁴
	4/1/4	2.500	0.625	2.500	3.125	1.250	1.2x10 ⁻⁴
	1/1/1	2.500	2.500	2.500	1.250	1.250	3.3x10 ⁻⁴
	1/1.5/1	2.500	3.750	2.500	0.000	1.250	1.1x10 ⁻³

Figure S62 suggests that there is a positive effect of H₂ and CO partial pressure on propanal on both catalysts. In addition, it can be confirmed by means of the H₂/CO/C₂H₄ ratio that when there is a low ratio of either H₂ or CO, the rate to propanal is also low (Table S12). According to these results we can suggest that the reaction mechanism on both catalysts is probably the same, but it may be that the rate determining step (rds) changes (i.e., the number of species involved changes) for each active site (Rh³⁺ and Rh⁺), and/or that these sites modify the adsorption energies of the intermediates involved in the propanal formation pathway, which causes that the apparent activation energy of the reaction changed between catalysts. In particular, the apparent orders of positive CO and with very similar values between both catalysts suggest that CO participates in the rds of the reaction mechanism on both active sites, with moderate CO adsorption (i.e., similar CO adsorption enthalpies) preventing deactivation by site poisoning. Meanwhile, the positive contribution of H₂ and the apparent order variation from 1.44 (Rh-MFI-cal) to 1.09 (Rh-MFI-calred) suggest that the rds in both catalysts is not the same; in Rh-MFI-calred participates at least two H* while in Rh-MFI-cal involved at least three H* adsorbed. Furthermore, a different rds supports the observed differences in the apparent activation energies determined for each active site (Table S10).

In most studies in the literature (*ACS Catal.*, 11, 2021, 14575; *J. Am. Chem. Soc.*, 145, 2023, 2911-2929; *ACS Catal.*, 7, 2017, 3584-3590), a negative order in CO is observed on Rh based catalysts, which has been explained by the strong interaction of Rh⁺ with CO, blocking the active sites and decreasing the reaction rate. Nevertheless, in contrast to those studies, a positive order in CO (ca. 0.6) and low apparent activation energies (~20 KJ/mol), has been reported in (*Nature* 609, 2022, 287) for a Rh-WO_x/Al₂O₃ sample containing Rh³⁺ sites, similar as it occurs in our case.

Thus, for comparative purpose, the effect of the partial pressure of CO has been also studied in a Rh impregnated sample over amorphous SiO₂ (Aerosil 200, Rh/SiO₂) with 0.1wt% Rh loading, ensuring a good dispersion of Rh over the support. Operando IR-MS studies shows the formation of Rh⁺ species in the Rh/SiO₂ calcined sample, and the contribution of small Rh⁰ nanoparticles in the Rh/SiO₂ calcined reduced sample. Notice that Rh³⁺ is not detected in any catalyst.

Figure S64 shows the effect of the partial pressure of CO and H₂ on the rates of propanal formation on the Rh/SiO₂-cal (violet line) and Rh/SiO₂-calred (red line) samples.

FigureS64. Reaction orders for 0.1wt% Rh/SiO₂-cal (violet) and 0.1wt% Rh/SiO₂-calred (red) catalysts at 70 °C. Rates for hydroformylation to propanal formation as a function of (a) H₂ partial pressure ($P_{\text{CO}} = 2.5$ bar, $P_{\text{C}_2\text{H}_4} = 2.5$ bar) and (b) CO partial pressure ($P_{\text{H}_2} = 2.5$ bar, $P_{\text{C}_2\text{H}_4} = 2.5$ bar) for a total operating pressure of 10 bar.

In addition, the influence of the CO and H₂ partial pressure, and different ratios of H₂/CO/C₂H₄ on the formation rate to propanal is displayed in Table S13

Table S13. Effect of the partial pressure of H₂ and CO, and the ratio CO/H₂/C₂H₄ on the formation rate of propanal (r, mol_{Propanal}/g_{cat}·h) in the hydroformylation of ethylene on 0.1wt% Rh/SiO₂-cal and 0.1wt% Rh/SiO₂-calred catalysts. Reaction conditions: 70 °C, 10 bar and gas mixture flow 80 mL/min.

Catalyst	H ₂ /CO/C ₂ H ₄	Partial Pressure (bar)					r _{propanal} (mol _{Propanal} /g _{cat} ·h)
		P _{CO}	P _{H₂}	P _{C₂H₄}	P _{Ar}	P _{N₂}	
0.1wt% Rh/SiO₂- cal	1/4/4	0.625	2.500	2.500	3.125	1.250	1.0x10 ⁻⁴
	1/1/1	2.500	2.500	2.500	1.250	1.250	3.3x10 ⁻⁵
	1.5/1/1	3.750	2.500	2.500	0.000	1.250	3.0x10 ⁻⁵
	4/1/4	2.500	0.625	2.500	3.125	1.250	1.1x10 ⁻⁵
	1/1/1	2.500	2.500	2.500	1.250	1.250	2.9x10 ⁻⁵
	1/1.5/1	2.500	3.750	2.500	0.000	1.250	4.3x10 ⁻⁵
0.1wt% Rh/SiO₂- calred	1/4/4	0.625	2.500	2.500	3.125	1.250	2.9x10 ⁻⁵
	1/1/1	2.500	2.500	2.500	1.250	1.250	2.7x10 ⁻⁵
	1.5/1/1	3.750	2.500	2.500	0.000	1.250	2.7x10 ⁻⁵
	4/1/4	2.500	0.625	2.500	3.125	1.250	7.5x10 ⁻⁶
	1/1/1	2.500	2.500	2.500	1.250	1.250	3.0x10 ⁻⁵
	1/1.5/1	2.500	3.750	2.500	0.000	1.250	4.6x10 ⁻⁵

The ca. 0 order on the 0.1wt% Rh/SiO₂-calred catalyst and the negative order for the 0.1wt% Rh/SiO₂-cal catalyst suggest that active sites formed during reaction from calcined catalysts, i.e., Rh⁺ easily adsorbs CO in a stable way (Rh(I)(CO)₂), which can obstruct ethylene adsorption at the same site, thus the site is poisoned by carbonyls. Therefore, we have observed in the case of the Rh/SiO₂ cal catalyst that higher CO coverage has a negative effect on ethylene hydroformylation, as has been previously reported in the literature (*ACS Catal.*, 11, 2021, 14575; *J. Am. Chem. Soc.*, 145, 2023, 2911-2929; *ACS Catal.*, 7, 2017, 3584-3590). However, when the calcined catalyst is reduced (0.1wt% Rh/SiO₂-calred), these sites change on the surface, forming mostly Rh⁰ nanoparticles that do not strongly interact with CO, showing almost zero order to CO. Meanwhile, the H₂ in both cases has a positive contribution. The apparent orders suggest that at least 1 or 2 adsorbed H* participate in the rds, as in the Rh-MFI-calred sample.

When we compare the results obtained on Rh-MFI and Rh/SiO₂, we can say that the local environment of Rh sites and their chemical properties has a strong influence on the catalytic and kinetic behaviour. Thus, the Lewis acid properties of metal cations can be tuned due to confinement effects, altering their coordination degree and electrophilicity, which in the case of zeolites is different compared to the same cation in other supports. This affects their interaction with reactants and intermediate species and in definitive have an effect on their catalytic performance.

We have included some comments in the manuscript (Page 14) and a detail analysis of the reaction orders and apparent activation energies are included in the SI (Section 4 entitled Kinetic studies in ethylene hydroformylation).

PAGE 14:

In addition to this dynamic catalyst behaviour and the stabilization of high oxidation state Rh³⁺ sites, the zeolite plays an important role in the reaction mechanism as extracted from kinetic studies of the reaction orders in CO and H₂ for propanal formation and the apparent activation energies of propanal formation. Kinetic studies are done at low temperature (50-70 °C) and under differential reaction conditions, i.e in the kinetic regime (details in SI section 4). As mentioned previously, the propanal apparent activation energies for the Rh-MFI samples are in all cases lower than those generally reported in the literature (*J. Am. Chem. Soc.* 2023, 145, 2911-2929) for

impregnated Rh based catalysts (see Table S10). This can be explained by the confinement interaction between the metal site and intermediate species and the zeolite framework (*Nat. Commun.* 2022, 13:821), influencing the adsorption enthalpies of intermediate products (more discussion in SI of section 4). Supporting this idea, the calculated reaction orders in CO and H₂ for propanal formation differ between the Rh-MFI samples and a reference Rh/SiO₂ impregnate sample. In the case of Rh-MFI catalysts, a positive effect of CO partial pressure on propanal formation is observed independent on the catalyst pre-treatment (see Figure S62), while it is practically zero or slightly negative in the Rh/SiO₂ sample (more details in SI, Figure S64), this last value in line with literature data (*ACS. Catal.*, 11, 2021, 14575; *J. Am. Chem. Soc.*, 145, 2023, 2911-2929; *ACS Catal.*, 7, 2017, 3584-3590). This may indicate different interaction strength of CO with Rh species, which depends not only on the chemical state but also on the local environment of the metal site. On the other hand, the H₂ order is slightly different among the catalysts, from close to 1.5 in the Rh-MFI-cal, to 1 in the Rh-MFI and Rh/SiO₂ samples, indicating differences in the H* coordination involvement in the rate determining step for propanal formation (more discussion in SI).

Comments:

Considering a molecular mechanism at solid's interface, can the authors comment on the role of the support as well as on the dissociation mechanism (energy?) of the Rh₂O₃ to Rh(3+) under reductive atmosphere?

Answers: Experimentally, the disruption of Rh₂O₃ into single site can be followed in the operando IR-MS studies. In fact, the formation of CO₂ (m/z = 44) and H₂O (m/z = 18) in the initial state of the reaction at 50 °C once the pressure has reached 10 bar, is clearly seen by mass spectrometry in the Rh-MFI calcined sample (Figure S20a and b), while not observed in the Rh-MFI calcined reduced sample (Figure S20c) nor Rh/MFI impregnated calcined reduced sample (Figure S20d). The formation of CO₂ and H₂O may come from the reactant induced fragmentation of Rh₂O₃ clusters into single sites. Notice that propanal is immediately detected by mass spectrometry and is also detected in the IR spectra, indicating a fast fragmentation and interaction of Rh sites with the

reactants to generate the in situ Rh(CO)L intermediate specie involved in propanal formation.

According to the referee, we have incorporated this discussion in the SI (Page 27).

“The disruption of Rh₂O₃ to Rh³⁺ species and/or to the intermediate stable Rh(CO)L specie can be followed in the operando IR-MS studies. In fact, the formation of CO₂ (m/z = 44) and H₂O (m/z=18) in the initial state of the reaction at 50 °C once the pressure has reached 10 bar, is clearly seen by mass spectrometry in the Rh-MFI-cal sample (Figure S20a and b), while not observed in the Rh-MFI-calred sample (Figure S20c) nor Rh/MFI impregnated calcined reduced sample (Figure S20d). The formation of CO₂ and H₂O may come from the reactant induced fragmentation of Rh₂O₃ clusters into single sites. Notice that propanal is immediately detected by mass spectrometry and is also detected in the IR spectra, indicating a fast fragmentation and interaction of Rh sites with the reactants to generate the in situ Rh(CO)L intermediate specie involved in propanal formation”.

Additionally, H₂O and CO₂ are also detected in the mass spectra at higher temperature (135 °C), paralleling the formation of ethane, and being much more intense in the Rh-MFI-cal sample. This may come from the reaction of CO and H₂ with OH or O groups of the zeolite. Because the simultaneous formation of ethane which indicate sintering of single sites into metal clusters, we suggest that those O or OH groups have been participated in the stabilization of the Rh³⁺ site. Hence, from this result we can conclude in a strong interaction and stabilization of Rh³⁺ sites by silanols groups of the zeolite.

Figure S20: Mass spectra monitoring the evolution of CO₂ (red line) and H₂O (violet line) during operando IR-MS conditions at different reaction stages: i) 50 °C and 1bar, ii) pressure increase at 50 °C from 1 to 10 bar, iii) stabilization at 10 bar and 50 °C and further increasing temperature up to 200 °C, for (a) Rh-MFI cal (b) inset of Rh-MFI-cal market with a circle, (c) Rh-MFI-calred and (d) Rh/MFI impregnated samples. Other products displayed in the mass spectra are ethane, shown as a blue line and propanal as a brown line. Notice that H₂O is detected in all samples at 50 °C and 1bar which may come from the zeolite.

Regarding on the role of the support, on one hand, as said before, the role of the zeolite may be the stabilization of Rh³⁺ sites. This behaviour is not observed on a Rh/SiO₂ impregnated sample. Indeed, operando IR-MS studies done on the Rh/SiO₂ calcined sample doesn't show the formation of Rh³⁺ species, while only Rh⁺ is detected (Figure S63). On the other hand, the zeolite has a strong influence on modulating the electronic

properties of the metal sites, in modulating the Rh-CO adsorption enthalpy and in the stabilization of intermediate species as deduced from kinetic studies.

This has been discussed in the new version of the manuscript (Page 14).

PAGE 14: “In addition to this dynamic catalyst behaviour and the stabilization of high oxidation state Rh³⁺ sites, the zeolite plays an important role in the reaction mechanism as extracted from kinetic studies of the reaction orders in CO and H₂ for propanal formation and the apparent activation energies of propanal formation. Kinetic studies are done at low temperature (50-70 °C) and under differential reaction conditions, i.e in the kinetic regime (details in SI section 4). As mentioned previously, the propanal apparent activation energies for the Rh-MFI samples are in all cases lower than those generally reported in the literature (J. Am. Chem. Soc. 2023, 145, 2911-2929) for impregnated Rh based catalysts (see Table S10). This can be explained by the confinement interaction between the metal site and intermediate species and the zeolite framework (Nat. Commun. 2022, 13:821), influencing the adsorption enthalpies of intermediate products (more discussion in SI of section 4). Supporting this idea, the calculated reaction orders in CO and H₂ for propanal formation differ between the Rh-MFI samples and a reference Rh/SiO₂ impregnate sample. In the case of Rh-MFI catalysts a positive effect of CO partial pressure on propanal formation is observed independent on the catalyst pre-treatment (see Figure S62), while it is practically zero or slightly negative in the Rh/SiO₂ sample (more details in SI, Figure S64), this last value in line with literature data (ACS. Catal., 11, 2021, 14575; J. Am. Chem. Soc., 145, 2023, 2911-2929; ACS Catal., 7, 2017, 3584-3590). This may indicate different interaction strength of CO with Rh species, which depends not only on the chemical state but also on the local environment of the metal site. On the other hand, the H₂ order is slightly different among the catalysts, from close to 1.5 in the Rh-MFI-cal to 1 in the Rh-MFI and Rh/SiO₂ samples, indicating differences in the H coordination involvement in the rate determining step for propanal formation (more discussion in SI)”.

Comments:

The scheme 1 and 2 are unfortunately not readable and more detailed mechanism for the first step should be postulated.

Answers:

Following the referee's suggestion, we have included new schemas.

Schema 1

Schema 2

Schema 3

Schema S2

Comments:

Finally, in the case of the P-doped catalyst, what is the nature of the Rh-P linkage (and Rh:P ratio in single site) in the “stabilized” catalyst?

Answers: As comment above, we propose a Rh-O=P- or Rh-O(H)-P interaction instead of Rh-P. The Rh-P interaction has been excluded in our study based on ^{31}P NMR and XANES data. The XANES spectra of the Rh-P-MFI calred sample at the P K-edge show that the P oxidation state is 5+, which is in line with ^{31}P NMR data, confirming the existence of phosphate species. Unfortunately, EXAFS at soft X-ray energies like P K-edge (2.146 KeV) is very challenging, especially under operando reaction conditions. Due to the low P concentration in the sample, the signal-to-noise ratio didn't give EXAFS with good statistics to be analyzed and therefore the Rh: (O) P ratio in the stabilized catalyst is difficult to asses.

We have included a comment in page 57 of the revised version of the SI “Due to the low P concentration in the sample, the signal-to-noise ratio didn't give EXAFS with good statistics to be analyzed and therefore the Rh: (O) P ratio in the stabilized catalyst is difficult to asses.”

Comments:

A comparison of the authors reported strategy and the encapsulation of commercially available Rh(H)(CO)(PPH₃)₃ should be enlightening.

Answers: We agree with the referee that encapsulation of a commercial Rh(H)(CO)(PPH₃)₃ complex in the MFI zeolite matrix would be very interesting. Indeed, homogeneous Rh phosphine complexes are known to be very active and its encapsulation inside the zeolite matrix would be even more interesting affording high stability. Nevertheless, we do not see the way to encapsulate the Rh(H)(CO)(PPH₃)₃ complex in the MFI zeolite either by direct synthesis or by post-synthesis.

Comments:

In conclusion, the novelty of this study lies in the understanding of catalyst nature and evolution under reaction conditions which however needs to be more detailed. Also, the use of heterogenized single-site Rh catalyst for ethylene hydroformylation has been reported several times earlier, with also much higher activity and ethylene is not a challenging substrate for this reaction.

Accordingly, due to questionable novelty, I cannot recommend the manuscript to be published in Nature Communications in its present form. Major revisions addressing at least the comments above are required.

Answers: Probably the interest of this work has not been clearly exposed. In this work a novel synthesis methodology is reported which allows the in situ confinement of phosphate ions inside the zeolite channels for the stabilization of metal single sites. The interest of this novel synthesis methodology is demonstrated in this work in the hydroformylation of olefins used as model reaction and using Rh@MFI zeolite as reference catalyst. Accordingly, we performed a detail fundamental study of the evolution of active Rh sites under hydroformylation conditions, getting fundamental insights into the nature of active sites and its deactivation trend under high temperature reaction conditions and highlighting the role of phosphate ions in this context.

We have modified partially the text in the new version of the manuscript in order to highlight the impact of our research.

PAGE 4-5

“Here a novel synthesis strategy allowing the stabilization under hydroformylation reaction conditions of active Rh species in absence of phosphine ligands is presented. In a first part of the work, a detailed analysis of the nature of active Rh species, their evolution and stabilization under hydroformylation working conditions is performed, using a Rh MFI zeolite prepared by a one pot hydrothermal method and submitted to different thermal treatments. Operando Infrared spectroscopy coupled with mass spectrometry (IR-MS) is combined with high-angle annular dark field imaging and scanning transmission electron microscopy (HAADF-STEM), X- ray adsorption spectroscopy (XAS) and Infrared (IR) spectroscopy of CO as probe molecule and validated with kinetic and catalytic studies in order to track the dynamic trend of the catalytic active specie. Thus, it is shown that the initial Rh⁰ and Rh₂O₃ clusters, present in the reduced (Rh⁰@MFI) and calcined (Rh₂O₃@MFI) catalysts respectively, disrupt under reaction conditions in different oxidized single site Rh species, Rh(I)(CO)₂ and high oxidation state Rh³⁺, with the last one acting as a more effective precursor for low temperature ethylene hydroformylation with syngas to produce propanal. The resultant Rh-MFI calcined catalyst is more active compared to state-of-the-art phosphine free

solid catalysts when operating at low temperature (50-100 °C), with TOF of 99 h⁻¹ at 90 °C and ~92 % selectivity to the aldehyde. Additional to these results, in a second part of our work, we present the possibility to stabilize under reaction conditions the single Rh³⁺ active site, even under energetically favoured metal sintering conditions, by developing a novel synthesis strategy in which phosphorous is introduced within the zeolite channels by using tetraethylphosphonium hydroxide as template. After H₂ activation it generate in situ phosphorous species, stabilizing Rh³⁺ sites and promoting propanal formation. The high steric hindrance of the zeolite channels promote regioselectivity when using propylene as substrate, opening new perspectives in the design of regioselective catalysts.

We also thanks the referee for his/her valuable comments, improving the quality of the work and giving additional information of active sites and reaction mechanism.

Reviewer #3 (Remarks to the Author):

Comments:

This work conducted by Zhao et al. involves the synthesis of Rh-MFI catalysts using a one-pot method. These catalysts exhibit high selectivity for the industrially-important hydroformylation reaction at low temperatures (50°C). The addition of phosphate species as a modification shows promise in enhancing the catalyst's stability. Characterization of the catalysts was performed using XAS, IR, and STEM techniques. However, several important questions need to be addressed before considering acceptance in Nat. Commun. This work might be better suited for a specialized catalysis journal.

1. The key novelty lies in the high selectivity of propanal at low temperatures due to the presence of Rh³⁺ species. However, the conversion remains very low (e.g., <1% at 50°C), and the Rh³⁺ species are prone to clustering despite the introduction of P species. At 200°C, the selectivity of hydroformylation drops to approximately 20%, which is not notably impressive compared to previously reported values. Additionally, it is not appropriate to compare the selectivity (Fig 3b), with values reported in the literature

when there are significant differences in conversion. Conversion was defined in Methods but it was not reported in this work.

Answers: In the present study, catalytic tests have been done under differential conditions, i.e., in the kinetic regime, linking operando XAS and IR-MS studies with catalytic studies done in a fixed-bed reactor, in order to obtain mechanistic information of active Rh sites, placing special attention on the intrinsic activity of active sites and their stability. We used a gas hour space velocity of 8000 h⁻¹, resulting in ethylene conversion below 1% in the 50-150 °C temperature range. Under these conditions, and in order to compare with literature data, where a variety of reaction conditions are used, the space time yield of propanal production (STY, mmol/g_{cat}.h) and the intrinsic activity of the active site (TOF, h⁻¹) are used in our manuscript.

Following the referee suggestion, new tables have been incorporated in the SI, including the values of conversion and selectivity to propanal on the most representative Rh-MFI catalysts.

Table S1. Ethylene conversion and selectivity to propanal on the Rh-MFI calcined sample.

Contact time (mg·min·ml ⁻¹)	GHSV (h ⁻¹)	T (°C)	Conver sion(%)	Selectivity to propanal (%)	STY-Propanal (mmol·h ⁻¹ ·g _{cat} ⁻¹)	TOF (h ⁻¹)
2.86 ^a	8000	50	0.23	93.10	0.53	23.50
			0.29	94.33	0.67	30.10
2.86 ^a	8000	70	0.67	92.92	1.58	68.22
			0.46	95.04	1.06	47.84
2.86 ^a	8000	90	0.98	91.84	2.21	98.68
			0.65	93.59	1.50	66.90
2.86 ^a	8000	120 ^b	2.06	85.32	4.29	192.03
			0.54	94.01	1.24	55.51

2.86 ^a	8000	150	0.80	79.11	1.54	68.77
			0.49	78.70	0.94	42.28
2.86 ^a	8000	200	8.18	16.99	3.39	151.90
			6.40	18.04	2.82	126.13

^a Without colour are initial values, while those with a green background correspond to final values

^b Reaction duration was 12h at 120 °C, while at the others temperatures, 2-3h

Table S7. Ethylene conversion and selectivity to propanal on the Rh-MFI-calred sample.

Contact time (mg·min·ml ⁻¹)	GHSV (h ⁻¹)	T (°C)	Conver sion(%)	Selectivity to propanal (%)	STY-Propanal (mmol·h ⁻¹ ·g _{cat} ⁻¹)	TOF (h ⁻¹)
2.86 ^a	8000	50	0.32	90.77	0.68	30.46
			0.16	94.90	0.36	15.99
2.86 ^a	8000	70	0.41	95.08	0.92	41.35
			0.27	94.74	0.61	27.45
2.86 ^a	8000	90	0.51	95.46	1.16	51.92
			0.58	95.77	1.31	58.77
2.86 ^a	8000	120 ^b	1.49	83.58	2.95	132.13
			0.63	92.32	1.38	61.54
2.86 ^a	8000	150	1.13	76.21	2.05	91.74
			0.86	61.88	1.27	56.84
2.86 ^a	8000	200	10.86	21.20	5.48	245.14
			7.51	21.81	3.89	174.27

^a Without colour are initial values, while those with a green background correspond to final values

^b Reaction duration was 12h at 120 °C, while at the others temperatures, 2-3h

Table S9. Ethylene conversion and selectivity to propanal on the Rh-(O)-P-MFI-calred sample

Contact time (mg·min·ml ⁻¹)	GHSV (h ⁻¹)	T (°C)	Conver sion(%)	Selectivity to propanal (%)	STY-Propanal (mmol·h ⁻¹ ·g _{cat} ⁻¹)	TOF (h ⁻¹)
2.86 ^a	8000	50	0.52	95.10	1.21	45.94
			0.24	98.32	0.59	22.33
2.86 ^a	8000	70	0.71	95.35	1.64	62.63
			0.38	97.69	0.90	34.42
2.86 ^a	8000	90	0.80	94.51	1.84	70.00
			0.75	95.89	1.74	66.43
2.86 ^a	8000	120 ^b	2.06	84.54	4.23	161.32
			0.81	89.69	1.76	67.14
2.86 ^a	8000	150	1.71	65.36	2.72	103.62
			1.30	54.75	1.73	65.98
2.86 ^a	8000	200	13.66	21.00	6.99	266.34
			11.12	22.01	5.96	227.28

^a Without colour are initial values, while those with a green background correspond to final values

^b Reaction duration was 12h at 120 °C, while at the others temperatures, 2-3h

Nevertheless, according to the suggestion of the referee, we did additional experiments decreasing the gas hour space velocity from 8000 to 800 h⁻¹, in order to increase the conversion of ethylene (Figure S8). As can be seen the selectivity to propanal remains above 80% in the 50-120 °C temperature range, with conversions up to 7%.

Figure S8. Variation of the selectivity to propanal with the conversion of ethylene on the Rh-MFI-cal sample at 50 °C, 70 °C and 90 °C. Reaction conditions: Ethylene: CO: H₂: N₂= 1: 1: 1: 0.5, 10 bar, GSHV from 8000 to 800 h⁻¹.

For comparative propose, in Table S3, we have included the data of the most active catalysts reported in the literature in the gas phase ethylene hydroformylation reaction. In this table, the conversion and selectivity to propanal are given at different temperatures and contact times.

Table S3 Conversion of ethylene and selectivity to propanal on the most active state-of-the-art phosphine free and phosphine based catalysts in ethylene hydroformylation.

Catalyst	WHSV (mL/g·h)	GHSV (h ⁻¹)	T (°C)	P (bar)	Conversion (%)	Selectivity (%)	TOF (h ⁻¹)	Ref ^c
Rh/MCM-41	3600	-	200	1	24.5	9.9	156	23
Rh ₁ Co ₁ /MCM-41	3600	-	200	1	41.7	23.8	210	23
Rh ₁ Co ₃ /MCM-41	3600	-	200	1	52.4	16.5	192	23
Rh-Co/Al ₂ O ₃	50000	-	200	20	-	19.6	94.5	24
Rh-V/SiO ₂	3600	3600	115	1	0.2	7	0.24	11
Rh/SiO ₂	6000	-	180	10	4	45	42	9
0.6Rh0.23Co/SiO ₂	6000	-	180	10	~9	52	42	9
Rh-0.7W/Al ₂ O ₃	9230 2142	-	100	10	10 27	96 96	-	14
RhCo ₃ /MCM-41	3600	-	160	1	17.7	52.5	~130	21
Rh/2.9ReO _x -Al ₂ O ₃	46000	-	150	1	-	45	~3.6	10
DPPPTS-RhAl ₁ /SiO ₂ ^b	-	2000	120	10	-	-	134	25
Rh-Dpppe/SiO ₂ ^b	6000	-	140	10	-	97	1172	26
Rh-Xantphos-SILP/NC ^b	7400	-	120	8	~70	95	600	27
Rh/POL-PPh ₃ ^b	-	2000	120	10	96.2	96.1	4530	28
Rh/POL-PPh ₃ ^b	-	5000	120	10	88.8	95.4	10373	28
Rh ₂ P/SiO ₂ ^b	-	2773	200	50	53	75	~190	8

^a With a green background correspond to reactions at 200 °C.

^b With a brown background correspond to phosphine based catalysts.

^c references are included in the SI

Table S1, S3, S7 and S9 and Figure S8 have been included in the new version of the SI.

A second remark of the referee is the selectivity to propanal at 200 °C, which drop to ~20%. Similar results are also found in the literature reporting selectivity's between 10 and 32% at 200 °C as seen in Table S3. However, when operating below 200 °C; the selectivity to propanal is in most cases above 70%, similar to that of our work. The reason to include the data at 200 °C in our work is to highlight the role of phosphate groups in stabilizing Rh³⁺ species, which otherwise are unstable at higher temperatures. This allows for higher propanal yields as indicated in Figure 6.

Finally, we agree with the referee that comparing the selectivity with values reported in the literature when there are significant differences in conversion is not correct. Figure 3b is a 3D representation including TOF, selectivity and temperatures. Notice that TOF is directly related to conversion. Therefore, we consider the figure representative of the state of the art including conversion and selectivity data at different temperatures. Therefore, we will keep it in the manuscript. Notice that this figure includes only data of phosphine free solid catalysts, which has been indicated in the figure caption of the new version of the manuscript.

Comments:

2. The apparent activation barriers for hydroformylation appear to be very low (e.g., 19 KJ/mol). Although these barriers were measured at low conversion with minimal transport limitations, the catalyst deactivation observed at higher temperatures would reduce the measured barriers. Thus, the intrinsic barriers of hydroformylation here should be higher than the measured values, which raises questions when using them for comparison with reported data.

Answers: We agree with the referee that 19 KJ/mol is very low compared to the state-of-the-art (usual values are between 42-72 KJ/mol). Because that, we took special care to make sure that we were not controlled by either external or internal diffusion (Figure S1). In addition, the analysis of apparent activation energies for propanal have been done in the 50-90 °C temperature range, conditions at which the Rh-MFI-cal catalyst remain stable. Therefore, we believe that the value is correct. Notice that a similar value

of 20 KJ/mol has been reported by the group of Christopher et al (*Nature* 2022; DOI: 10.1038/s41586-022-05075-4) on a Rhodium-WO_x catalyst with 0.23wt% Rh and 0.7wt% W. It is interesting to remark that Rh³⁺ has been observed in this catalyst, while no comment on the role of Rh³⁺ has been given by the authors. Meanwhile, apparent activation energies of 27 KJ/mol has been also reported by Arai et al. in a Rh/NaY zeolite (*J. Catal.*, 1982, 75(1), 188-189, [https://doi.org/10.1016/0021-9517\(82\)90134-8](https://doi.org/10.1016/0021-9517(82)90134-8)).

A comment has been included in the new version of the manuscript in Page 10. “Interestingly, a similar value of 20 KJ/mol) has been recently reported by Christopher et al for a 0.23 wt% Rh-0.7wt% WO_x /Al₂O₃ catalyst containing Rh³⁺ sites (*Nature* 2022:DOI: 10.1038/s41586-022-05075-4) “.

On the other hand, the referee needs to take into account that we are given apparent activation energies. Therefore, if the adsorption enthalpies of intermediate products is strong, it lead to a decrease in the apparent E_a value. In the case of metal confined in zeolites, it has been shown that the confinement interaction between the metal site and the zeolite framework has a significant impact on both the stabilization of the metal species and on the catalytic reactivity (*Nature Communications* 2022, 13:821). Also stabilization of the transition state by dispersion forces within the pores of the zeolite has been described (*ACS. Catal.*, 2019, 9(2), 1539-1548).

Based on the effect of the partial pressure of CO and H₂ on the propanal formation rate on both Rh-MFI and Rh/SiO₂ impregnated samples, this work clearly shows how the local environment of Rh sites and their chemical properties has a strong influence on the catalytic and kinetic behaviour. (See discussion in section 4 of the SI and in Page 14 of the manuscript)

Page 14. “In addition to this dynamic catalyst behaviour and the stabilization of high oxidation state Rh³⁺ sites, the zeolite plays an important role in the reaction mechanism as extracted from kinetic studies of the reaction orders in CO and H₂ for propanal formation and the apparent activation energies of propanal formation. Kinetic studies are done at low temperature (50-70 °C) and under differential reaction conditions, i.e in the kinetic regime (details in SI section 4). As mentioned previously, the propanal apparent activation energies for the Rh-MFI samples are in all cases generally lower than those reported in the literature (*J. Am. Chem. Soc.*, 2023, 145, 2911-2929) for

impregnated Rh based catalysts (see Table S10). This can be explained by the confinement interaction between the metal site and intermediate species and the zeolite framework (Nat. Commun., 2022, 13:821), influencing the adsorption enthalpies of intermediate products (more discussion in SI of section 4). Supporting this idea, the calculated reaction orders in CO and H₂ for propanal formation differ between the Rh-MFI samples and a reference Rh/SiO₂ impregnate sample. In the case of Rh-MFI catalysts a positive effect of CO partial pressure on propanal formation is observed independent on the catalyst pre-treatment (see Figure S62), while it is practically zero or slightly negative in the Rh/SiO₂ sample (more details in SI, Figure S64), this last value in line with literature data (ACS. Catal., 11, 2021, 14575; J. Am. Chem. Soc., 145, 2023, 2911-2929; ACS Catal., 7, 2017, 3584-3590). This may indicate different interaction strength of CO with Rh species, which depends not only on the chemical state but also on the local environment of the metal site. On the other hand the H₂ order is slightly different among the catalysts, from close to 1.5 in the Rh-MFI-cal to 1 in the Rh-MFI-calcred and Rh/SiO₂ samples, indicating differences in the H* coordination involvement in the rate determining step for propanal formation (more discussion in SI).”

Comments:

3. The assignment of peaks corresponding to Rh-C and Rh-O bonds in EXAFS is questionable. The indicated positions of these two bonds suggest a bond length difference of ~0.5 Å, which is unlikely in the co-presence of Rh-C bonds from CO adsorption and Rh-O bonds in RhO_x moieties (or with the support). Instead, the apparent two peaks between 1-2 Å (e.g., Fig. S9) are typically the result of strong destructive interference between Rh-C and Rh-O bonds with similar distances and scattering characteristics. The EXAFS fitting should be added and the co-linear scattering of Rh-C=O at a higher shell should also be included, and EXAFS plots in both k and R spaces should be presented. Additionally, it is recommended to perform wavelet-transformed EXAFS analysis to differentiate the presence of Rh-Rh bonds at positions similar to those of the Rh-C=O scattering pathway.

Answers: We thank the referee for the careful reading and the comments. We added the corresponding fits showing the Rh-O and Rh-C distances at ~2.03 Å and ~1.84 Å,

respectively in the Supplementary Information. As per the reviewer's comments co-linear scattering of the Rh-C=O bond were carried out. Moreover, we added in the Figure 2 caption the following sentence: "The arrows indicating the different contributions are only indicative. The EXAFS fitting results are reported in the Supplementary Information." Moreover, the wavelet-transformed EXAFS analysis has been added in the main manuscript.

Comments:

4. The authors deduce small particle size based on peak intensities in FT-EXAFS, yet, which may not necessarily hold true, as peak intensity can also be affected by structural disordering in cluster morphologies. Furthermore, the peak intensity between 1-2 Å is lower than that of Rh₂O₃. Does this imply that Rh^{δ+} is under-coordinated? EXAFS fitting should be performed for the fresh, activated, and reaction conditions.

Answers: We agree with the referee that the FT peak intensity depends on the disorder and the coordination number and that to disentangle the two parameters is not obvious and often complementary information is needed. In the present case the particle size has been characterized by HAADF-STEM. The here detected reduction in intensity of the FT higher shell feature around 2.2-3.1 Å, corresponding to the Rh-Rh scattering contribution of the Rh₂O₃ system, most likely corresponds to small particle size, in agreement with the HAADF-STEM measurements. Instead, the lower peak intensity between 1-2 Å with respect to that of Rh₂O₃ could correspond to under-coordination or higher local disorder, but to disentangle these two parameters is here more difficult. The fitting of the data does show however a loss of longer range structure beyond ~3 Å together with a disordered, low degeneracy scattering between 2.2-3.1 Å indicative of small disordered particles consistent with the HAADF-STEM data. The fitting results have been added to the Supplementary Information.

Comments:

5. The authors state that Rh is in a high oxidation state based on its similar edge position to that of Rh₂O₃. However, it is not necessarily true, as Rh(I) complex species can also exhibit a similar edge position to Rh₂O₃. For a better comparison, the XANES spectrum of Rh(I) complex standards, such as those with a local group of Rh(I)(CO)_xO₂, should

be included. Moreover, the presence of Rh-O bonds in EXAFS is not exclusively indicative of the presence of Rh³⁺ since isolated Rh⁺ species on the support (e.g., Rh(I)(CO)₂O₂ with a square-planar geometry) can also exhibit Rh-O bonds.

Answers: We thank the referee for the suggestion. Unfortunately, we did not measure the suggested Rh(I) references. We agree that the exploitation of the Rh(I)(CO)_xO₂ reference spectra could permit to access quantitative results. Indeed, the formation of Rh-CO/Rh(0) at the expenses of the Rh₂O₃ phase can be only qualitatively followed in Figure 2c. In the literature we didn't find the measurement of the suggested spectrum, which has been instead calculated by Hülsey et al. (<https://doi.org/10.1038/s41467-019-09188-9>). Effectively, in such calculations the edge position of the of Rh(I)(CO)_xO₂ seems close to the ones of the Rh³⁺ species, but still slightly shifted toward lower energies. In the absence of empirical data, we can't exclude the presence of reduced species, but we believe that the here investigated samples show the dominant presence of Rh centers having a high effective oxidation state, in agreement also with the IR data. We added a figure in the supplementary material reporting additional Rh 2+ and 3+ references measured in the present study and we carefully revised the manuscript to avoid ambiguities.

Comments:

6. In the schema, it was proposed that the monocarbonyl Rh(CO)L complex interacts with OH groups of the zeolite. The OH groups seem to be restored after depressurization and purging with Ar (Fig. S12b). Can it be due to the typical trace amount of water in Argon rather than the dis-interaction with Rh(CO)L?

Answers: Thanks for the comment. According to the referee suggestion, we did a blank experiment on pure MFI zeolite using the same experimental protocol, and in this case, we only observed a small increase of the IR band at 3703 cm⁻¹ due to silanol groups which are regenerated in the presence of trace amounts of water (see Figure S18, blue line). Accordingly, in the Rh-MFI cal sample, the peak at ~3400 cm⁻¹ (see Figure S18, red line) which re-appear after partial desorption of propanal can be safely assigned to OH groups (silanol nest or hydrogen bonded OH) (*Micropor Mesopor Mat.*, 288, 2019, 109582) regenerated after the desorption of propanal.

We have included this figure in the new version of the SI labelled as Figure S18. We have also included a small comment in Page 26 of the SI: “The contribution of trace amounts of water in the Ar flow in the restoring of the OH groups can be discarded as shown in Figure S18”.

Figure S18. IR spectra of pure MFI zeolite (blue line) compared to that of the Rh-MFI-cal sample (red line) under Argon flow after de-pressurization (10 bar → 1bar) of the spectroscopic cell at the reaction temperature of 90 °C.

Comments:

7. The function by which P stabilizes the Rh sites, whether through electronic, geometric, or steric effects, remains unclear.

Answers: As commented in the manuscript, we couldn't find electronic differences in the nature of Rh species between the Rh-MFI-calred and Rh-(O)-P-MFI-calred samples using IR-CO (Figure S55) as probe molecule, and neither morphological differences could be found by STEM (Figure S48-51). However, from in situ XAS and IR-CO we observed the stabilization of Rh³⁺ species during the oxidative disruption of Rh clusters which takes place under reaction conditions. We propose that Rh³⁺ ions are in situ stabilized via Rh-O=P or Rh-O(H)-P interaction, indicating a geometric rather than electronic effect of phosphate ions in the stabilization of Rh³⁺.

We have included some comments in the new version of the manuscript (Page 17)

Page 17, “Based on our results, it is believed that Rh^{3+} is stabilized via Rh-O=P or Rh-O(H)-P interaction, (see Schema 3)”.

Comments:

8. Abbreviations such as XAS, HAADF, STEM, IR, etc., should be defined upon their first mention. Grammar issues should be addressed approximately. For the sake of readiness, the colors for different samples should be kept consistent such as avoiding the reversal seen in Figs. 4a and 4b.

Answers: We have included the definition of each technique in Page 5. We also modified the colours of Figure 4a and 4b.

Reviewer #4 (Remarks to the Author):

Comments:

In this work, Concepcion, Corma and co-workers report the preparation and mechanistic investigations of highly efficient Rh-MFI catalyst for hydroformylation. The authors also show a one-pot strategy for the preparation of Rh-P-MFI catalyst, in which the oxidized Rh can be stabilized by phosphate ions within the zeolite pores. Spectroscopic tools used in this study further provide insights into the dynamic behaviors of the catalysts. Overall, the new catalysts, preparation strategy and mechanistic investigation in this work present significant findings for the field of catalysis. However, there are some major issues that need to be addressed before the work can be published in Nature Communication:

1. The dynamic behavior proposed in Scheme 1 (and Scheme 3) needs further discussion. The mechanism in which the reaction initiated with a Rh^{3+} species (the second species) seems not follow the classic Heck and Breslow mechanism. Further investigations such as the detection of CO_2 generation at the beginning of

hydroformylation and the effect of CO partial pressure on the catalyst activity, could give more convincing conclusions.

Answers: According to the suggestion, we have performed additional experiments studying the effect of the CO partial pressure on the catalyst activity. A detailed discussion has been included in response to point 7 of referee 2, and new kinetic data have been included in section 4 of the SI.

We also analysed the disruption of Rh_2O_3 to single sites in the initial state of the reaction at 50 °C and 10 bar, following the formation of CO_2 and H_2O by mass spectrometry (MS) in the operando IR studies. A detailed discussion has been included in response to point 8 of referee 2, and new data regarding the disruption of Rh_2O_3 have been included in the SI.

In the context of the reaction mechanism, computational methods will be very valuable and may provide further insights into the reaction path, but it is out of the scope of our work. In any case, we can give some comments in this respect. Initiation of the classic Heck and Breslow mechanism requires CO desorption from $\text{Rh}^+(\text{CO})_2$. This is not required when starting from Rh_2O_3 , where a $\text{Rh}(\text{CO})\text{L}$ intermediate specie is formed under reaction conditions by disruption of Rh_2O_3 and simultaneous coordination of one CO molecule and ethylene molecule. The low CO interaction, preventing deactivation by poisoning the active site, is demonstrated by the positive CO reaction order. However, further studies are required for a detailed mechanistic study.

Nevertheless, in this work we settled that the zeolite plays a critical role in the reaction mechanism. This has been extracted from kinetic studies of the reaction orders in CO and H_2 for propanal formation, from the apparent activation energies of propanal formation, and from additional experiments done on a Rh impregnated sample (Rh/SiO_2). A detailed discussion of these features is included in the SI and in the response to referee 2.

Comments:

2. The apparent activation energies are quite low (19.5 KJ/mol for Rh-MFI-cal). Is it possible that the reaction proceeded with diffusion-controlled kinetics?

Answers: As commented in the Supporting Information we are working under conditions of no transport limitations (see Figure S1). A detailed discussion about the apparent activation energies has already been included in response to point 2 of referee 3.

Comments:

3. It is confusing to me that the E_a for ethane formation is comparable and even smaller than propanal formation (Figure S7) at 50-90 °C, whereas the selectivity to aldehyde is >95%. In addition, the E_a values of ethane measured at 423-473 K are reported in Table S1 and the main article. The authors should comment on their choice of E_a values.

Answers: In the case of ethane, a change in the apparent activation energy is observed when moving from low temperature (50-120 °C) to high temperature (120-200 °C) (as shown for instance in Figure S7). This is due to a change in the nature of active sites as demonstrated from operando IR and XAS studies and supported by STEM, where initially single sites with some contribution of ultra-small clusters predominate, whereas at higher temperature they tend to aggregate into bigger metal clusters. In the literature most studies operate at high temperature and under those conditions the activation energies are close to ours. For example reported values are: 60-121 KJ/mol for Rh/SiO₂ (*Ind. Eng. Chem. Res.*, 2020, 59 (42) 18771-18780); 134-167 KJ/mol for Rh-Zn/SiO₂ (*J. Am. Chem. Soc.*, 1985, 107(24), 7216-7218); 82-110 KJ/mol for Rh/Al₂O₃ (*ACS. Catal.*, 2019, 9(12), 10899-10912; *J. Catal.*, 1984, 85(1), 89-97); 90 KJ/mol for Rh/NaY (*J. Catal.*, 1982, 75(1), 188-189) and 102-109 KJ/mol for Rh-ReO_x/Al₂O₃ (*ACS. Catal.*, 2019, 9(12), 10899-10912). Due to comparative propose, we included in Table S10 the apparent activation energy for ethane in the 120-200 °C range. Nevertheless, in agreement to the referee we have included a new column in Table S10 of the SI giving the apparent activation energy for ethane also in the 50-120 °C range.

Table S10. Apparent activation energies according to the Arrhenius plot based on reaction rates of ethylene hydroformylation to propanal and ethylene hydrogenation to ethane, in the temperature range 50-90°C for propanal and 50-120°C and 120-200°C for ethane.

Sample	E _a Propanal (kJ/mol)	E _a Ethane (kJ/mol)	
		50-120°C	120-200°C
Rh-MFI-cal	19.5 ± 1.5	18.5 ± 0.8	131.4 ± 0.2
Rh-MFI-calred	31.6 ± 2.1	32.2 ± 0.1	95.5 ± 0.5
Rh-(O)-P-MFI-calred	26.5 ± 2.2	45.1 ± 1.5	89.2 ± 0.6
Rh/MFI-calred	61.3 ± 3.2	22.4 ± 0.5	87.8 ± 1.0

^a The calculation of E_a are based on final STY values of different temperature.

In addition, we included a comment in Page 16: “The calculated apparent activation energy in the 120-200 °C temperature range for the hydrogenation reaction (89.2-131.4 KJ/mol) are higher than that for the hydroformylation reaction (19.5-31.6 KJ/mol)^{23, 36, 40}, in line with literature data and with the increase of ethane formation with increasing temperature.”

Regarding the second question, why the selectivity to propanal is > 95% on the Rh-MFI-cal sample when operating at low temperature, despite their similar apparent E_a for ethane formation (18.5 KJ/mol) than that of propanal formation (19.5 KJ/mol), it has to be considered that we are given apparent activation energy values. As indicated, the apparent E_a = E_{a,rds} + ∑ΔH^o_{ads}, where ΔH^o_{ads} represents the enthalpy of adsorption in the adsorption equilibrium constant and E_{a,rds} is the activation energy of the rate determining step (rds). Therefore, we can have different adsorption enthalpies of intermediate products and different activation energies (E_{a,rds}), compensating each other in a way that the apparent activation energy is the same.

Comments:

4. The authors should provide more procedure details of the catalyst test. For example, which time period is used for the calculation of STY?

Answers: STY values are given close to steady state conditions. Nevertheless, we have included new tables in the SI, labelled as Tables S1, S7 and S9 for the R-MFI cal, Rh-MFI calred and Rh-(O)-P-MFI calred samples respectively, where the STY are included

at each temperature highlighting the initial value and the last value after ~ 2-3 h of operation.

Comments:

There are also other minor points:

1. In Figure 5d, Rh-MFI-cal is less reactive than Rh-MFI-calred, whereas in the main article Rh-MFI-cal is more active.

Answers: The referee is right. However, it has to be taken into account the different reaction temperature of Figure 5d (200 °C) compared to the rest of the paper. Rh-MFI-cal sample is less stable operating at high temperature compared to the Rh-MFI calred one, due to the different thermal stability of active species (Rh^{3+} and $\text{Rh}^+(\text{CO})_2$).

If we base ourselves on the information of apparent activation energies in Table S10, it would be expected that catalyst Rh-MFI-cal would be more reactive than the rest because it has the lowest apparent activation energy, but due to the instability of the active site at high temperatures, the number of catalyst sites decreases significantly (sintering effect), decreasing the yield to propanal. However, notice that addition of phosphate ions allows the stabilization of the Rh^{3+} species even at high temperatures, preserving a higher number of active sites that allow a significant increase in the formation of propanal.

Comments:

2. It is recommended to cite some representative review articles for hydroformylation: e.g., Chem Rev 2012, 5675; Green Synth. Catal. 2021, 247

Answers: We thanks the referee for the comment. We have included those references in the revised version of the manuscript.

Comments:

3. The author should further check their references. For example, ref 34 and 36 are the same.

Answers: We thanks the referee for this remark. We have corrected it, and also new references have been included.

Reviewer #5 (Remarks to the Author):

Comments:

In this article, Zhao et al. reported low temperature hydroformylation of ethylene heterogeneously catalyzed by Rh single sites stabilized by phosphorous which is confined inside MFI type zeolite. The high activity of the Rh catalyst is ascribed to the disruption of Rh₂O₃ clusters to the Rh single site. Interestingly, the Rh₀ clusters do not form those active species and hence less active. The authors characterized the catalysts with very high level sophisticated techniques and conducted operando studies to prove whether the true active site is single site or not. There has been quite some discussion on this topic quite recently (doi.org/10.1021/acs.chemrev.2c00495) whether single site remain as it is or it forms something else due to high surface energy and XAS seems to be a valuable technique to track the dynamic evolution of the active site. Discussing such literature within this manuscript will also send a general message to the community that in-situ spectroscopy is highly important for further development of this field. Nevertheless, XAS also has limitations (doi.org/10.1021/acscatal.3c01116), which should also be discussed and hence complementary techniques such as FTIR makes further progress in the field. Hence, this work makes these points clearer.

Answers: We appreciate very much the comment of the referee. This work represents an example where the combination of various spectroscopic techniques working under operando conditions is very important for the in-depth characterization of active species. It is also important to add the own inherent limitation of each technique.

In line with the referee remark, we have included a comment in Page 3 highlighting the limitations of the techniques and the need of complementary techniques.

Page 3 “Additionally, because each technique has its own inherent limitations, complementary techniques are very necessary to prove the structure of active sites (ACS Catal 2023, 13, 9, 6462-6473; Chem Rev 2023, 123, 379-444).”

Comments:

Overall, the authors did a very elaborative study from catalyst synthesis, reaction kinetics, operando investigation and supported their hypothesis with strong evidences. The study is well presented with a huge amount of data collected. I however have few points of concern which might helps in improving the manuscript and have better understanding from a point of view of a common reader and should be carefully reviewed before this manuscript can be accepted in final form.

1.It is quite unclear what is the role of phosphorous as it is highly important as most of the industrial catalysts (molecular Rh complex) requires P in the first coordination shell. Since the authors put it in the title as Rh-P-MFI zeolite and consistently showed that the first shell is Rh-C or Rh-O and this is why I am confused of the need of P and the title can be mis-leading. It will be nice to elaborate and show distinct evidence that there is no P in the first coordination sphere. The presented ^{31}P NMR and the P K-edge XAS only give information about presence of P.

Answers: As well indicated by the referee, EXAFS data shows C/O in the first coordination shell, and from ^{31}P NMR we can exclude a Rh-P interaction. In this context, it is true that the label Rh-P-MFI is confusing. Therefore, in order to avoid confusion to the reader we have change “Rh-P-MFI” by “Rh-(O)-P-MFI” along the manuscript.

Comments:

2. Looking at the IR data, there is a consistent CO vibrational frequency in the range 2040-2060 cm^{-1} which I believe is CO adsorbed over Rh cluster. This vibrational frequency is also observed in the IR-CO at -65 °C in Figure 1e. Therefore, it is clever to assume that there is always a fraction of Rh cluster present in the catalyst which I guess the authors also agrees. Hence the authors should strongly state (especially in the conclusion part) that the possibility of ultra small Rh clusters cannot be completely ruled out. Or if the authors disagree with the statement, then should provide strong evidences.

Answers: The referee is right, we cannot discard the coexistence of ultra small Rh clusters. This has already been mentioned along the manuscript, but not clearly

specified in the conclusion. According to the referee we have add a comment in the conclusion part of the revised version of the manuscript.

Page 21, “In both cases, beside the predominance of single sites, the co-existence of ultra small Rh clusters cannot be discarded”.

Comments:

3. In the XAS spectra, is it correct that the scattering around 2.5 Å is assigned to Rh-Rh or Rh-C≡O? I doubt that Rh-C≡O will show such a strong scattering due to much distortion than the Rh-Rh. The authors should validate this claim by measuring Rh carbonyl complexes with XAS.

Answers: We thank the reviewer for his attention to detail. While indeed in cases where the peak at 2.5 Å is particularly pronounced most of the intensity may be attributed as coming for Rh-Rh interactions, we have now also performed EXAFS analysis and show that the Rh-Rh scattering overlaps with the resulting patterns of co-linear Rh-CO scattering giving rise to the resulting pattern in the 2.0-3.0 Å range of the FT EXAFS signal. As before the labels try to emphasize the presence of both species.

Comments:

4. Even though the authors relate the CO vibrational frequency at 2156 cm⁻¹ to the Rh³⁺, in such high frequency many reports were claiming to have weakly absorbed CO on the support or gas phase CO. Do the authors conducted any blank experiment to check CO adsorption over the zeolite without any Rh? Especially there is also a shoulder peak between 2150-2200 cm⁻¹ which is not explained. Further the ratio of the symmetric to antisymmetric peak at 2112 and 2030 cm⁻¹ do not seems to follow a trend. I would assume that this ratio should be constant over the time of CO dosing. In all other spectra reported in the electronic supporting information, there is a distinct peak around 2030 cm⁻¹. Is there any reason for that?

Answers: As indicated by the referee, the IR band at 2156 cm⁻¹ can be also associated to silanol groups and the one at 2138 cm⁻¹ to physisorbed CO in the zeolite channels, overlapping the IR bands of Rh³⁺-CO. In order to determine the contribution of silanols

and physisorbed CO in the IR spectra of the Rh containing sample, we did a blank experiment on the pure zeolite. As shown in Figure S21, zeolite contribution is minimal (violet line), which is expected given that CO adsorption was carried out at $-65\text{ }^{\circ}\text{C}$, rather than the usual $-170\text{ }^{\circ}\text{C}$, required to titrate silanol groups.

We have included a comment in the Figure 1e caption:

“The contribution of CO coordinated to silanol groups (2156 cm^{-1}) and physisorbed CO (2135 cm^{-1}) is minimal, as determined from a blank experiment done on pure MFI (see Figure S21)”.

And the IR spectra of CO adsorption on the pure MFI zeolite is included in the SI (Figure S21).

Figure S21. IR-CO at $-65\text{ }^{\circ}\text{C}$ and at saturation CO coverage (2 mbar) for Rh-MFI-cal (red line) and pure MFI (violet line) samples, in order to present the contribution of CO coordinated to silanol groups (2156 cm^{-1}) and physisorbed CO (2135 cm^{-1}) overlapped by the Rh^{3+} -CO band

Regarding the shoulder peak between $2150\text{--}2200\text{ cm}^{-1}$ observed in Figure 1e, that is due to the high CO dosing, which always generate a tail at the high and low frequency region.

On the other hand, the referee is completely right that ratio of the symmetric to antisymmetric CO vibrations of Rh(CO)₂ (i.e. at 2100-2090 and 2035-2025 cm⁻¹, respectively) do not seem to follow a trend among the different samples and reaction conditions. The referee has to consider that the ratio of the integrated asymmetric and symmetric peaks areas is linked to the angle between carbonyl groups in the dicarbonyl compound as shown in Equation 1. (*Nature Communication* 2021,12: 4698; *J. Chem. Phys.* 70, 1979, 1219-1224). Thus, It can be expected that the geometry of the Rh(CO)₂ complex, which is closely associated with the coordination environment, may change under vacuum or reaction conditions.

$$\text{Equation 1 } A \frac{A_{\text{asym}}}{A_{\text{sym}}} = \tan^2(\alpha)$$

Finally, regarding the following comment done by the referee: “In all other spectra reported in the electronic supporting information, there is a distinct peak around 2030 cm⁻¹”. We don’t understand exactly the question of the reviewer, but it is true that in Figure S24, a shoulder in the low frequency side of the 2034 cm⁻¹ IR peak (around 2016 cm⁻¹) is observed.

This shoulder, located at around 2003-1981 cm⁻¹, can be ascribed to CO interacting with under-coordinated sites of Rh⁰, or due to background changes. Change in the background of the IR spectra could also result in some asymmetries in the low frequency range.

Comments:

Other comments:

1. In scheme 1, 2 and 3, some of the features are very difficult to read. I suggest either less information or bigger font size. For example the way it is shown in Page 20 of supplementary information is much clearer. The same goes to Figure 1 (c) and (d). X-axis values need larger font size.

Answers: We have modified it

Comments:

2. In page 9, line 175: worth noting instead of worthnothing

Answers: We have corrected it in the new version of the manuscript.

Comments:

3. It is not clear what the authors want to describe in the sentence 254 (page 9) “tendency also observed in the operando XAS studies”. What tendency they refer to?

Answers: we referred the loss of propanal selectivity due to metal sintering, which is observed by operando IR, HTEM and XAS. All spectroscopies follows the same trend.

Comments:

4. Page 17 (line 254) of supporting information, k range is represented as Å⁻¹ instead of Å.

Answers: Thanks for the comment . We have modified it in the new version of the manuscript.

REVIEWER COMMENTS

Reviewer #2 (Remarks to the Author):

In their revised manuscript (NCOMMS-23-26412A), Zhao et al. provide complete and detailed answers to all reviewers comments.

One can appreciate the great efforts made to provide additional insights into the scientific discussions around the catalytic activity against various alkenes and on the mechanism at play here, as well as the accurate modifications in the text.

The authors performed numerous additional experiments with great scientific rigor and also a complete analysis and discussion of the new results presented.

Compared to the initial submission, the manuscript is greatly improved and gives a very comprehensive view of the zeolite-encapsulated Rh catalysis reported.

Thus, I recommend the publication of this manuscript in Nature Communications.

Reviewer #3 (Remarks to the Author):

The reviewer acknowledges the authors' efforts to address the concerns raised. However, there remain some critical questions that require further clarification:

1. The role of P species at elevated temperatures (e.g., 200°C) remains uncertain and confusing. On one hand, the authors assert that Rh is present in Rh₀ clusters, while on the other hand, in the Response file, they claim that P can stabilize Rh₃₊ (“...the data at 200 °C in our work is to highlight the role of phosphate groups in stabilizing Rh₃₊ species...”). Moreover, the enhanced stability by P should be clarified in relation to their own catalysts free of P or previously reported catalysts since stable conversion can be easily obtained at 200C using Rh bi/metallic catalysts. This should be clearly stated.

2. The EXAFS fittings are questionable and may suffer from over-fitting issues. Most of the R factors range between 0.03 and 0.04, which is huge, casting doubt on the reliability of the fitting results. The fitting methods are also unclear and confusing. More comprehensive explanations of these methods are necessary.

3. The low activation barrier still appears to be affected by deactivation effects. Despite the authors' assertion that measurements were conducted at a stable conversion rate between 50-90°C, there is clear declining conversion at both 50°C and 90°C (Fig. 3a). So deactivation must contribute to the observed barrier. Can the authors consider measuring the barrier using a single catalyst, starting at 90°C for a few hours, decreasing to 50°C, and then increasing it back to 90°C? This approach might help partially compensate for the deactivation effect by averaging TOFs at the same temperature.

4. Comparing selectivity solely in terms of TOF may not be equitable and could occasionally be misleading. While TOF is correlated with conversion, precisely determining TOF can be challenging, as it typically depends on the quantification of active sites, which can vary among research groups employing different methods such as chemisorption or ICP (for single atoms). Additionally, the stoichiometric ratios of probe adsorbate to metal can be complex, particularly when both linear and gem dicarbonyl species are present. Thus, it would be beneficial to add another figure (selectivity vs. conversion) as Fig. 3c to show where the current work is located.

Reviewer #4 (Remarks to the Author):

In the revised manuscript, the authors have been able to address all the concerns raised by this reviewer. The quality of the manuscript has been increased with all the additional experiments suggested by this reviewer and by the other reviewer. This reviewer recommends the acceptance of the manuscript in its present form.

Reviewer #5 (Remarks to the Author):

The authors have clarified all my concerns and included in the revised manuscript. Even though the role of phosphate is not clearly understood as also other reviewers raised the same, future work in this line will certainly help in unravelling the catalyst structure. I recommend for publication.

The authors appreciate the reviewer's effort and time for the critical review of the article. The manuscript has been updated with new information and a point to point answer to all the comments risen by reviewers is included in this document. The answer to the reviewers is indicated in red.

All changes done in the manuscript are marked in yellow in the revised version.

Reviewer #2 (Remarks to the Author):

In their revised manuscript (NCOMMS-23-26412A), Zhao et al. provide complete and detailed answers to all reviewers comments.

One can appreciate the great efforts made to provide additional insights into the scientific discussions around the catalytic activity against various alkenes and on the mechanism at play here, as well as the accurate modifications in the text. The authors performed numerous additional experiments with great scientific rigor and also a complete analysis and discussion of the new results presented. Compared to the initial submission, the manuscript is greatly improved and gives a very comprehensive view of the zeolite-encapsulated Rh catalysis reported. Thus, I recommend the publication of this manuscript in Nature Communications.

Answers: We thanks the referee for his/her positive comments.

Reviewer #3 (Remarks to the Author):

The reviewer acknowledges the authors' efforts to address the concerns raised. However, there remain some critical questions that require further clarification:

1. 1. The role of P species at elevated temperatures (e.g., 200 °C) remains uncertain and confusing.

Answers: Rh metal clusters are initially present in both (promoted and un-promoted) calcined reduced samples as shown from STEM and IR-CO, and in both cases an oxidative disruption of the Rh clusters takes place under reaction conditions. However, from in situ XAS and IR-CO we observed in the P promoted sample, the stabilization of Rh³⁺ species during the oxidative disruption of Rh clusters. We proposed that Rh³⁺ ions are in situ stabilized via Rh-O=P or Rh-O(H)-P interaction while direct Rh-P interaction has been excluded in our study based on ³¹P NMR and XANES data.

This has been reported in the manuscript in PAGE 18: ... In both samples the Rh metal cluster present in the calcined-reduced samples in the presence of syngas are partially disrupted to form Rh-CO bonds, but differently to the Rh-MFI sample, in the Rh-(O)-P-MFI catalyst a higher amount of isolated Rh³⁺ sites are formed under working conditions.

1.2. On one hand, the authors assert that Rh is present in Rh⁰ clusters, while on the other hand, in the Response file, they claim that P can stabilize Rh³⁺ (“...the data at 200 °C in our work is to highlight the role of phosphate groups in stabilizing Rh³⁺ species...”). Moreover, the enhanced stability by P should be clarified in relation to their own catalysts free of P or previously reported catalysts since stable conversion can be easily obtained at 200 °C using Rh bi/metallic catalysts. This should be clearly stated.

Answers: The stabilization of Rh³⁺ species in the Rh-(O)-P-MFI-calred sample, results in higher propanal formation, which is clearly observed in Figure 5a in the low temperature range (50-90 °C), and also at higher temperature (200 °C) as shown in Figure 6. In both figures, the catalytic activity is compared to their own P free sample.

Figure 5a. Promoting effect of P in the catalytic performance.

Figure 6. Catalytic performance at 200 °C and 10 bar on the Rh-MFI-cal (blue), Rh-MFI-calred (violet) and Rh-(O)-P-MFI-calred (red) samples.

This is reported in our manuscript in PAGE 18: ..“The promoting effect of high oxidised Rh single site (i.e. Rh³⁺) is confirmed in the catalytic studies, resulting in higher propanal formation. Thus at 90 °C the propanal yield increase from 1.3 up to 1.8 mmol_{propanal}/g_{cat}·h in the presence of P (Figure 5a), behaviour also confirmed by operando IR-MS studies (Figure S63).”

PAGE 19: ...“The stabilization effect of oxidised Rh species by P is more evident at 200 °C, conditions where Rh single sites are demonstrated to be unstable, and tend to agglomerate with a corresponding loss of propanal activity. In particular, at those conditions, propanal yields of 7.0 mmol_{propanal}/g_{cat}·h are obtained in the P-doped Rh-MFI sample, being ~1.3 and ~2.1 times higher than in the un-doped calcined and calcined reduced samples (Figure 6).”

Concerning the other comment of the reviewer to compare our catalyst with other Rh bi/ metallic samples, we followed his/her advise and in Table S3 we have included the data of the most active catalysts reported in the literature in the gas phase ethylene hydroformylation reaction. According to this table, bimetallic samples give similar selectivities as compared to our samples. On the other hand, conversions around 50% are given for the bimetallic Rh₁Co₃/MCM-41 samples, that if expressed in a molar base per gram of catalyst result in 19.3 mmol_{C₂H₄ converted}/g_{cat}·h and 3.2 mmol_{propanal formed}/g_{cat}·h. Notice that these values are lower than the 33.3 mmol_{C₂H₄ converted}/g_{cat}·h and 7.0 mmol_{propanal formed}/g_{cat}·h obtained with our Rh-(O)-P-MFI sample. Regarding stability, the Rh-(O)-P-MFI sample shows a slightly higher catalyst deactivation, see (Figure R1b) compared to the RhCo/MCM-41 catalysts reported in the literature (Figure R1a), but due to the different reaction conditions and weight hourly space velocity (WHSV) used in the literature compared to that of our work, a direct conclusion on deactivation differences cannot be raised.

Figure R1. A) Catalytic performance of MCM-41 supported RhCo catalysts from the literature (Nature Communications 2020, 11:1887); B) our catalyst. Reaction conditions: A) C₂H₄/CO/H₂/Ar = 3/3/3/3 mL/min, 200 mg, atmospheric pressure, WHSV= 3600 mL/g_h; B) C₂H₄/CO/H₂/N₂ = 10/10/10/5 mL/min, 100 mg, 10 bar pressure, WHSV= 21000 mL/g_h.

2. The EXAFS fittings are questionable and may suffer from over-fitting issues. Most of the R factors range between 0.03 and 0.04, which is huge, casting doubt on the reliability of the fitting results. The fitting methods are also unclear and confusing. More comprehensive explanations of these methods are necessary.

Answers: We thank the referee for asking a more detailed description since we believe the fitting results robust. Indeed, we believe important to specify that we have used 5-9 variables, when, considering the exploited k and R ranges, up to 12 degrees of freedom were available. A systematic approach was used in adding paths to exclude possible data overfitting. As additional variables were added, we ensured that the reduced χ^2 was minimized for the best fit (see: Bevington, P. R.; Robinson, D. K., *Data Reduction and Error Analysis for the Physical Sciences*. 2 ed.; McGraw-Hill: New York, 1992). This effect is illustrated in the corresponding tables summarised in table R1.

We do note however, that the fits presented were carried out to 4 Å instead than in the intended 1-3.2 Å range (corresponding to 12 degrees of freedom available for the fit). Indeed, in r-space the actual features being fit span the 1-3.2 Å range. We thank the referee for the careful reading and we updated the fit tables and statistics, and further proofed the EXAFS fit tables considering the 1-3.2 Å range for the fittings. These changes do not alter the conclusions or results of the paper but are showing lower R factors.

We have updated the Experimental section accordingly to better describe the fitting procedure.

Experimental section update in the SI, Page 5

In situ XAS spectroscopy.

X-ray Absorption spectra (XAS) of powder pellets were acquired at the Rh K-edge, where the powder material was diluted in boron nitride when necessary to ensure an absorption jump close to 1. Using a Si (311) double crystal monochromator, XAS spectra were collected in transmission mode for the reference samples and in fluorescence mode for the samples by means of a multichannel SDD fluorescence detector available at the CLÆSS beamline of the ALBA synchrotron¹. Several XAS repeats were collected to ensure reproducibility and statistics for the ex-situ measurements. Spectra processing and analysis was carried out with the Athena software package². The energy scale was calibrated by setting the first inflection point of the Rh metal foil taken as 23220.0 eV. EXAFS were extracted using the AUTOBK algorithm employing a spline in the k range of 0 to 15.9 Å⁻¹ for ex situ samples and 0 to 14 Å⁻¹ for operando samples, having a R_{bkg} of 1.1. The FEFF6 code^{3,4} was used for scattering path generation, with Rh-O and Rh-Rh paths being generated from the crystal structures of Rh₂O₃ and Rh⁰ and Rh-CO paths from model structure mp-683938 from the Materials Project^{5,6}. Multi (k^1, k^2, k^3)-weighted fits of the data were carried out in r -space over r - (1-3.2 Å) and k - ranges as indicated in the Supporting Information. The S_0^2 value was set to 0.9, and a global E_0 was employed with the initial E_0 value set to the first inflection point of the rising edge and the initial change in energy ΔE set to 0.0. Single scattering paths were fit in terms of a Δr_{eff} and σ^2 , which represent the deviation from the expected interatomic distances and the structural disorder, respectively with initial values of 0.0 Å and 0.003 Å². A systematic approach was used in adding paths to exclude possible data overfitting. As additional variables were added, we ensured that the reduced χ^2 was minimized for the best fit. Co-linear Rh-CO paths were fit with a common Δr_{eff} and σ^2 and consist of 4 paths Rh-C, Rh-O, Rh-O-C, Rh-C-O-C. Up to 9 variables have been considered in the fits, when, considering the exploited k and R ranges, up to 12 degrees of freedom were available. To assess the goodness of the fits both the R_{factor} (%R) and the reduced χ^2 (χ^2_{v}) were minimized⁷. Best fit models were determined using a grid search with fixed values for path coordination numbers (N) by employing Larch, the Python implementation of Artemis⁸.

Reference

1. Simonelli L, *et al.* CLÆSS: The hard X-ray absorption beamline of the ALBA CELLS synchrotron. *Cogent Phys.* **3**, 1231987 (2016).
2. Ravel B, Newville M. ATHENA, ARTEMIS, HEPHAESTUS: data analysis for X-ray absorption spectroscopy using IFEFFIT. *J. Synchrotron Radiat.* **12**, 537-541 (2005).
3. Newville M. EXAFS analysis using FEFF and FEFFIT. *J. Synchrotron Radiat.* **8**, 96-100 (2001).
4. Rehr JJ, Albers RC. Theoretical approaches to X-ray absorption fine structure. *Rev. Mod. Phys.* **72**, 621-654 (2000).
5. Wyckoff, R. W. G. Crystal Structures. Second Edition, *Interscience Publishers*, **1**, 7-83, New York (1963).

6. Jain A, et al. Commentary: The Materials Project: A materials genome approach to accelerating materials innovation. *APL Materials*. **1**, 011002 (2013).
7. Bevington, P. R., & Robinson, D. K. Data Reduction and Error Analysis for the Physical Sciences. Second edition, *McGraw-Hill*, New York (1992).
8. Newville M. Larch: An analysis package for XAFS and related spectroscopies. *J Phys Cond Ser*. **430**, 012007 (2013).

Table R1. EXAFS analysis where paths were added systematically to exclude possible data overfitting. Multi (k^1, k^2, k^3)-weighted fits carried out in r-space (1-3.2 Å) over a k-range of 3-12 Å⁻¹ using a Hannings window (dk 1), and $S_0^2 = 0.9$. Bond distances and disorder parameters (ΔE_{eff} and σ^2) were allowed to float having initial values of 0.0 Å and 0.003 Å² respectively, with a universal E_0 and $\Delta E_0 = 0$ eV.

Rh-MFI-cal

	R_{FACTOR}	0.025383	0.0334571	0.0845815
	X²_v	19.594064	21.451777	46.375652
	Var. No.	7	5	3
	S₀²	0.90	0.90	0.90
	ΔE₀ (eV)	2.3 (1.2)	2.5 (1.2)	2.7 (2.0)
Rh-C/N/O	N	3.8	3.8	3.5
	r (Å)	2.04 (0.01)	2.04 (0.01)	2.04 (0.02)
	σ² (x10³ Å²)	3.0 (0.5)	3.1 (1.2)	3.0 (0.1)
Rh-Rh	N	0.8		
	r (Å)	2.71 (0.04)		
	σ² (x10³ Å²)	8.8 (3.7)		
	N	1.5	1.5	
	r (Å)	3.09 (0.01)	3.09 (0.01)	
	σ² (x10³ Å²)	4.6 (1.3)	4.3 (1.2)	

Rh-MFI-cal Reaction feed 50 °C 10 bar

	R_{FACTOR}	0.0129275	0.0285713	0.0538785	0.0912033
	X²_v	1.6290385	2.3156759	3.2184118	4.3135943
	Var. No.	9	7	5	3
	S₀²	0.90	0.90	0.90	0.90
	ΔE₀ (eV)	-6.0 (2.6)	1.2 (2.2)	1.7 (2.5)	1.8 (2.7)
Rh-C/N/O	N	1.8			
	r (Å)	1.84 (0.03)			
	σ² (x10³ Å²)	3.0 (0.0)			

Rh-C/N/O	N	3.2	3.2	3.0	3.0
	r (Å)	2.00 (0.02)	2.04 (0.02)	2.04 (0.02)	2.04 (0.02)
	σ^2 (x10³ Å²)	3.3 (0.8)	6.8 (1.2)	5.6 (1.3)	5.6 (1.6)
Rh-Rh	N	0.5	0.8	0.8	
	r (Å)	2.61 (0.05)	2.67 (0.02)	2.69 (0.04)	
	σ^2 (x10³ Å²)	9.4 (3.3)	8.0 (2.2)	7.8 (3.8)	
	N	1.0	1.0		
	r (Å)	3.06 (0.02)	3.09 (0.03)		
Rh-O	N	1.0			
	r (Å)	2.99 (0.03)			
	σ^2 (x10³ Å²)	3.0 (0.0)			
Rh-O-C	N	1.0			
	r (Å)	2.99 (0.03)			
	σ^2 (x10³ Å²)	3.0 (0.0)			
Rh-C-O-C	N	0.5			
	r (Å)	2.99 (0.03)			
	σ^2 (x10³ Å²)	3.0 (0.0)			

Rh-MFI-cal Reaction feed 50 °C 1 bar

	R_{FACTOR}	0.0095307	0.0729816	0.0717561	0.1944336
	X²_v	0.5785378	3.1177155	2.3647858	5.2157064
	Var. No.	9	7	5	3
	S₀²	0.90	0.90	0.90	0.90
	ΔE_0 (eV)	-3.4 (2.1)	6.0 (3.4)	5.6 (2.9)	8.5 (3.9)
Rh-C/N/O	N	1.5			
	r (Å)	1.87 (0.02)			
	σ^2 (x10³ Å²)	3.0 (6.5)			
Rh-C/N/O	N	2.0	2.5	2.5	2.5
	r (Å)	2.03 (0.02)	2.08 (0.03)	2.08 (0.03)	2.10 (0.04)
	σ^2 (x10³ Å²)	4.1 (1.0)	9.9 (3.3)	10.0 (1.9)	10.0 (0.0)
Rh-Rh	N	0.8	1.0	1.0	
	r (Å)	2.74 (0.01)	2.76 (0.02)	2.76 (0.02)	
	σ^2 (x10³ Å²)	4.7 (1.3)	5.3 (1.9)	5.8 (1.8)	
	N	1.0	0.5		

	r (Å)	3.11 (0.02)	3.08 (0.40)		
	σ^2 (x10³ Å²)	6.4 (2.5)	25.2 (78.0)		
Rh-O	N	3.0			
	r (Å)	3.02 (0.02)			
	σ^2 (x10³ Å²)	3.0 (6.5)			
Rh-O-C	N	2.0			
	r (Å)	3.02 (0.02)			
	σ^2 (x10³ Å²)	3.0 (6.5)			
Rh-C-O-C	N	1.0			
	r (Å)	3.02 (0.02)			
	σ^2 (x10³ Å²)	3.0 (6.5)			

Rh-MFI-cal, reaction temperature 50 °C 114 min

	R_{FACTOR}	0.0130103	0.0243171	0.0429348	0.1434527
	X²_v	1.6107469	1.936348	2.5197571	6.6659265
	Var. No.	9	7	5	3
	S₀²	0.90	0.90	0.90	0.90
	ΔE_0 (eV)	-6.0 (2.9)	2.7 (2.2)	3.0 (2.1)	4.4 (3.5)
Rh-C/N/O	N	2.0			
	r (Å)	1.84 (0.02)			
	σ^2 (x10³ Å²)	3.0 (0.4)			
Rh-C/N/O	N	3.0	3.5	3.2	3.2
	r (Å)	2.00 (0.02)	2.05 (0.02)	2.05 (0.02)	2.05 (0.03)
	σ^2 (x10³ Å²)	3.4 (0.6)	10.0 (0.1)	9.1 (1.5)	8.5 (2.4)
Rh-Rh	N	0.8	0.5	0.5	
	r (Å)	2.61 (0.03)	2.64 (0.02)	2.65 (0.02)	
	σ^2 (x10³ Å²)	4.9 (2.3)	4.4 (2.0)	3.1 (1.6)	
	N	0.8	1.0	1.0	0.0
	r (Å)	2.73 (0.03)	2.75 (0.02)	2.76 (0.02)	2.80
	σ^2 (x10³ Å²)	4.9 (2.3)	4.4 (2.0)	3.1 (1.6)	
	N	1.0	1.0		
	r (Å)	3.06 (0.02)	3.11 (0.03)		
	σ^2 (x10³ Å²)	3.8 (1.2)	6.9 (2.5)		
Rh-O	N	0.2			
	r (Å)	2.88 (0.02)			

	σ^2 (x10 ³ Å ²)	3.0 (0.4)			
Rh-O-C	N	0.5			
	r (Å)	2.88 (0.02)			
	σ^2 (x10 ³ Å ²)	3.0 (0.4)			
Rh-C-O-C	N	0.2			
	r (Å)	2.88 (0.02)			
	σ^2 (x10 ³ Å ²)	3.0 (0.4)			

Rh-MFI-cal, reaction temperature 90 °C 107 min

	R_{FACTOR}	0.0178549	0.0580815	0.0887273	0.3210325
	X²_v	2.0076023	4.2003946	4.7291947	13.5482
	Var. No.	9	7	5	3
	S₀²	0.90	0.90	0.90	0.90
	ΔE₀ (eV)	-3.6 (3.2)	2.1 (2.5)	0.5 (2.7)	11.2 (5.3)
Rh-C/N/O	N	2.0			
	r (Å)	1.88 (0.03)			
	σ^2 (x10 ³ Å ²)	5.4 (3.2)			
Rh-C/N/O	N	2.0	2.0	2.0	2.5
	R (Å)	2.03 (0.04)	2.02 (0.03)	2.01 (0.03)	2.11 (0.06)
	σ^2 (x10 ³ Å ²)	6.7 (2.7)	10.0 (0.3)	10.0 (0.5)	10.0 (0.1)
Rh-Rh	N	2.0	3.0	2.0	
	r (Å)	2.71 (0.02)	2.75 (0.02)	2.75 (0.02)	
	σ^2 (x10 ³ Å ²)	6.1 (1.3)	9.2 (2.2)	6.0 (1.8)	
	N	0.8	1.0		
	r (Å)	2.82 (0.02)	2.87 (0.02)		
	σ^2 (x10 ³ Å ²)	6.1 (1.3)	9.2 (2.2)		
	N	1.5	1.5		
	r (Å)	3.06 (0.02)	3.05 (0.03)		
Rh-O	N	1.0			
	r (Å)	2.92 (0.03)			
	σ^2 (x10 ³ Å ²)	5.4 (3.2)			
Rh-O-C	N	2.0			
	r (Å)	2.92 (0.03)			
	σ^2 (x10 ³ Å ²)	5.4 (3.2)			

Rh-C-O-C	N	1.0		
	r (Å)	2.92 (0.03)		
	σ^2 (x10³ Å²)	5.4 (3.2)		

Rh-MFI-cal, reaction temperature 200 °C 238 min

	R_{FACTOR}	0.0221888	0.0748018	0.2144324
	X²_v	0.9157348	2.3815387	5.5570577
	Var. No.	7	5	3
	S₀²	0.90	0.90	0.90
	ΔE_0 (eV)	-3.0 (1.2)	-2.0 (1.5)	-3.3 (2.3)
Rh-C/N/O	N	2.0		
	r (Å)	1.89 (0.02)		
	σ^2 (x10³ Å²)	5.6 (2.1)		
Rh-C/N/O	N	1.0	1.2	
	r (Å)	2.07 (0.04)	1.96 (0.01)	
	σ^2 (x10³ Å²)	10.0 (0.2)	10.0 (14.5)	
Rh-Rh	N	2.0	3.0	3.0
	r (Å)	2.70 (0.01)	2.71 (0.01)	2.70 (0.02)
	σ^2 (x10³ Å²)	4.6 (0.7)	7.2 (1.1)	6.8 (1.5)
	N	1.0	1.5	1.5
	r (Å)	2.82 (0.01)	2.82 (0.01)	2.81 (0.02)
	σ^2 (x10³ Å²)	4.6 (0.7)	7.2 (1.1)	6.8 (1.5)
Rh-O	N	2.0		
	r (Å)	3.04 (0.02)		
	σ^2 (x10³ Å²)	5.6 (2.1)		
Rh-O-C	N	1.0		
	r (Å)	3.04 (0.02)		
	σ^2 (x10³ Å²)	5.6 (2.1)		
Rh-C-O-C	N	0.5		
	r (Å)	3.04 (0.02)		
	σ^2 (x10³ Å²)	5.6 (2.1)		

Rh-MFI-calred H₂ (158 min)

	R_{FACTOR}	0.0192593	0.0370501	0.0461089
	X²_v	1.0067229	1.427372	1.406485

	Var. No.	7	5	3
	S₀²	0.90	0.90	0.90
	ΔE₀ (eV)	-6.2 (1.0)	-6.6 (1.1)	-7.2 (1.0)
Rh-C/N/O	N	1.0		
	r (Å)	1.83 (0.03)		
	σ² (x10³ Å²)	5.7 (3.6)		
Rh-C/N/O	N	0.8	0.5	
	r (Å)	2.04 (0.03)	1.97 (0.07)	
	σ² (x10³ Å²)	3.3 (3.4)	14.9 (13.2)	
Rh-Rh	N	4.5	4.5	4.8
	r (Å)	2.65 (0.01)	2.64 (0.01)	2.64 (0.01)
	σ² (x10³ Å²)	10.0 (0.5)	10.0 (0.6)	10.0 (0.1)

Rh-MFI-calred reaction feed (162 min)

	R_{FACTOR}	0.0092958	0.0148526	0.0172712
	X²_v	0.6680807	0.7867227	0.7243424
	Var. No.	7	5	3
	S₀²	0.90	0.90	0.90
	ΔE₀ (eV)	-7.0 (0.6)	-7.2 (0.7)	-7.5 (0.6)
Rh-C/N/O	N	0.8		
	r (Å)	1.80 (0.05)		
	s² (x10³ Å²)	9.4 (4.4)		
Rh-C/N/O	N	0.8	0.2	
	r (Å)	2.03 (0.04)	2.00 (0.10)	
	s² (x10³ Å²)	12.4 (5.7)	15.0 (38.8)	
Rh-Rh	N	5.5	5.8	5.8
	r (Å)	2.64 (0.01)	2.64 (0.01)	2.64 (0.01)
	s² (x10³ Å²)	11.7 (0.4)	12.0 (0.4)	11.9 (0.4)

Rh-MFI-calred reaction feed (235 min)

	R_{FACTOR}	0.0129271	0.0675088	0.2431284
	X²_v	0.492889	1.9857287	5.8210852
	Var. No.	7	5	3
	S₀²	0.90	0.90	0.90
	ΔE₀ (eV)	-2.7 (0.9)	-1.8 (2.8)	-3.6

Rh-C/N/O	N	2.0		
	r (Å)	1.89 (0.02)		
	σ^2 (x10³ Å²)	8.2 (1.9)		
Rh-C/N/O	N	0.8	1.0	
	r (Å)	2.03 (0.02)	1.96 (0.03)	
	σ^2 (x10³ Å²)	3.5 (2.3)	9.6 (4.2)	
Rh-Rh	N	2.1	3.0	3.2
	r (Å)	2.71 (0.01)	2.71 (0.02)	2.71 (0.02)
	σ^2 (x10³ Å²)	6.9 (0.5)	9.1 (1.7)	9.0 (1.7)
	N	0.5	0.8	0.8
	r (Å)	2.82 (0.01)	2.82 (0.02)	2.82 (0.02)
	σ^2 (x10³ Å²)	6.9 (0.5)	9.1 (1.7)	9.0 (1.7)
Rh-O	N	2.0		
	r (Å)	3.04 (0.02)		
	σ^2 (x10³ Å²)	8.2 (1.9)		
Rh-O-C	N	1.0		
	r (Å)	3.04 (0.02)		
	σ^2 (x10³ Å²)	8.2 (1.9)		
Rh-C-O-C	N	0.5		
	r (Å)	3.04 (0.02)		
	σ^2 (x10³ Å²)	8.2 (1.9)		

Rh-MFI-calred reaction feed (339 min)

	R_{FACTOR}	0.0099122	0.0243035	0.0574369
	X²_v	1.2318101	2.2259711	4.1652724
	Var. No.	7	5	3
	S₀²	0.90	0.90	0.90
	ΔE_0 (eV)	-2.8 (0.9)	-3.7 (1.0)	-4.6 (1.2)
Rh-C/N/O	N	2.0		
	r (Å)	1.89 (0.02)		
	σ^2 (x10³ Å²)	5.9 (2.2)		
Rh-C/N/O	N	1.0	1.2	
	r (Å)	2.06 (0.02)	1.96 (0.03)	
	σ^2 (x10³ Å²)	3.5 (2.8)	10.0 (4.7)	
Rh-Rh	N	3.8	4.5	5.5

	r (Å)	2.71 (0.00)	2.71 (0.01)	2.70 (0.01)
	σ^2 (x10³ Å²)	4.5 (0.3)	5.2 (0.4)	6.3 (0.5)
Rh-O	N	1.0		
	r (Å)	3.04 (0.02)		
	σ^2 (x10³ Å²)	5.9 (2.2)		
Rh-O-C	N	2.0		
	r (Å)	3.04 (0.02)		
	σ^2 (x10³ Å²)	5.9 (2.2)		
Rh-C-O-C	N	1.0		
	r (Å)	3.04 (0.02)		
	σ^2 (x10³ Å²)	5.9 (2.2)		

Rh-(O)-P-MFI-calred H₂ (220 min)

	R_{FACTOR}	0.0148119	0.0278459
	X²_v	0.741241	1.1033472
	Var. No.	5	3
	S₀²	0.90	0.90
	ΔE_0 (eV)	-5.7 (0.7)	-6.5 (0.8)
Rh-C/N/O	N	0.8	
	r (Å)	1.99 (0.04)	
	σ^2 (x10³ Å²)	13.5 (5.6)	
Rh-Rh	N	5.5	5.8
	r (Å)	2.66 (0.01)	2.66 (0.01)
	σ^2 (x10³ Å²)	9.2 (0.4)	9.4 (0.4)

Rh-(O)-P-MFI-calred reaction feed (225 min)

	R_{FACTOR}	0.007952	0.0172773
	X²_v	0.6368189	1.0955189
	Var. No.	5	3
	S₀²	0.90	0.90
	ΔE_0 (eV)	-4.8 (0.5)	-5.4 (0.6)
Rh-C/N/O	N	0.8	
	r (Å)	1.98 (0.03)	
	σ^2 (x10³ Å²)	15.0 (0.5)	
Rh-Rh	N	5.5	5.8

r (Å)	2.68 (0.00)	2.68 (0.00)
σ^2 (x10³ Å²)	6.9 (0.2)	7.1 (0.3)

Rh-(O)-P-MFI-calred reaction feed (376 min)

	R_{FACTOR}	0.0083356	0.0117761	0.2040749
	X²_v	0.771595	0.8034054	11.023629
	Var. No.	7	5	3
	S₀²	0.90	0.90	0.90
	ΔE₀ (eV)	-2.3 (1.0)	-1.9 (0.7)	-4.8 (2.4)
Rh-C/N/O	N	1.0		
	r (Å)	1.85 (0.05)		
	σ^2 (x10³ Å²)	13.0 (2.0)		
Rh-C/N/O	N	1.8	1.8	
	r (Å)	2.00 (0.01)	1.99 (0.01)	
	σ^2 (x10³ Å²)	6.3 (1.7)	7.7 (1.3)	
Rh-Rh	N	2.8	3.0	4.0
	r (Å)	2.71 (0.01)	2.71 (0.00)	2.70 (0.02)
	σ^2 (x10³ Å²)	5.8 (0.3)	6.3 (0.3)	7.8 (1.3)
Rh-O	N	1.0		
	r (Å)	3.00 (0.05)		
	σ^2 (x10³ Å²)	13.0 (2.0)		
Rh-O-C	N	0.8		
	r (Å)	3.00 (0.05)		
	σ^2 (x10³ Å²)	13.0 (2.0)		
Rh-C-O-C	N	0.8		
	r (Å)	3.00 (0.05)		
	σ^2 (x10³ Å²)	13.0 (2.0)		

Rh-(O)-P-MFI-calred reaction feed (440 min)

	R_{FACTOR}	0.0121484	0.0182393	0.0335545
	X²_v	1.4833023	1.7324201	2.6079178
	Var. No.	7	5	3
	S₀²	0.90	0.90	0.90
	ΔE₀ (eV)	-4.7(1.0)	-4.8(0.8)	-5.5(0.9)
Rh-C/N/O	N	2.0		

	r (Å)	1.84(0.04)		
	σ^2 (x10³ Å²)	11.9(5.0)		
Rh-C/N/O	N	1.5	1.0	
	r (Å)	2.03(0.03)	1.98(0.04)	
	σ^2 (x10³ Å²)	10.0(2.1)	10.0(1.8)	
Rh-Rh	N	4.5	5.2	6.0
	r (Å)	2.69(0.00)	2.69(0.00)	2.69(0.01)
	σ^2 (x10³ Å²)	4.2(0.3)	4.9(0.3)	5.6(0.3)
Rh-O	N	1.0		
	r (Å)	2.99(0.04)		
	σ^2 (x10³ Å²)	11.9(5.0)		
Rh-O-C	N	2.0		
	r (Å)	2.99(0.04)		
	σ^2 (x10³ Å²)	11.9(5.0)		
Rh-C-O-C	N	1.0		
	r (Å)	2.99(0.04)		
	σ^2 (x10³ Å²)	11.9(5.0)		

3. The low activation barrier still appears to be affected by deactivation effects. Despite the authors' assertion that measurements were conducted at a stable conversion rate between 50-90°C, there is clear declining conversion at both 50°C and 90°C (Fig. 3a). So deactivation must contribute to the observed barrier. Can the authors consider measuring the barrier using a single catalyst, starting at 90 °C for a few hours, decreasing to 50 °C, and then increasing it back to 90 °C? This approach might help partially compensate for the deactivation effect by averaging TOFs at the same temperature.

Answers: Following the experiment proposed by the reviewer to compensate for deactivation effects, we have measured the activation barrier starting at 90 °C and maintained at that temperature for 2h, then decreasing the temperature to 70 and 50 °C, and increasing back to 70 and 90 °C. The reaction conditions are 10 bar, GHSV=8000 h⁻¹. The obtained activation barrier is 28.8 KJ/mol, compared to the initial one of 19.5 KJ/mol (see Figure R2). Nevertheless, the 28.8 KJ/mol is much lower than the values reported in literature (42-72 KJ/mol), confirming that the Rh-MFI cal sample gives much lower activation energy.

Figure R2. Calculation of apparent activation energies on the Rh-MFI cal sample, following the experimental conditions from low (50 °C) to high (90 °C) temperature (red) and vice-versa, from high (90 °C) to low (50 °C) and then back to 70 and 90 °C (violet).

In response to the referee, we have included a new Figure S8 in the SI (Page 12).

Figure S8. Arrhenius plot based on reaction rates of ethylene hydroformylation to propanal in the temperature range 50-90 °C, starting from 50 °C and then increasing to 70 and 90 °C (red) and starting at 90 °C, then decreasing to 70 and 50 °C and finally back to 70 and 90 °C in order to compensate for deactivation effects (violet). In both cases the values are very close in line with

the stability of the catalyst (E_a of 50-70-90 °C is based on final stabilized STY values, E_a of 90-70-50-70-90 °C is based on average STY values at different temperatures).

4. Comparing selectivity solely in terms of TOF may not be equitable and could occasionally be misleading. While TOF is correlated with conversion, precisely determining TOF can be challenging, as it typically depends on the quantification of active sites, which can vary among research groups employing different methods such as chemisorption or ICP (for single atoms). Additionally, the stoichiometric ratios of probe adsorbate to metal can be complex, particularly when both linear and gem dicarbonyl species are present. Thus, it would be beneficial to add another figure (selectivity vs. conversion) as Fig. 3c to show where the current work is located.

Answers: We agree with the referee that determining TOF values can be challenge, specifically at low metal loadings. However, it has to be taken into account that due to the different reaction conditions used in the literature, such as gas space velocity, total pressure and composition of the reactant feed, it is not appropriate to compare the level of conversion between different catalysts. We have included in Table S3 of the SI (Table R1), a comparison of conversion vs selectivity of state-of-the-art-phosphine free solid catalysts, which have been plotted in Figure R3. This figure has been added in the SI, as Figure S10.

Table R1. Conversion of ethylene and selectivity to propanal on the most active state-of-the-art phosphine free and phosphine based catalysts in ethylene hydroformylation.

Label on Fig. R3c	Catalyst	WHSV (mL/g·h)	GHSV (h ⁻¹)	T (°C)	P (bar)	Conversion (%)	Selectivity (%)	STY (mmol _{C₂H₄} converted/g _{cat} ·h)	STY (mmol _{propanal} formed/g _{cat} ·h)	TOF (h ⁻¹)	Ref
33	Rh/MCM-41	3600	-	200	1	24.5	9.9	9.0	0.9	156	23
34	Rh ₁ Co ₁ /MCM-41	3600	-	200	1	41.7	23.8	15.4	3.7	210	23
35	Rh ₁ Co ₃ /MCM-41	3600	-	200	1	52.4	16.5	19.3	3.2	192	23
36	Rh-Co/Al ₂ O ₃	50000	-	200	20	~6.0	19.6	-	-	94.5	24
37	Rh-(O)-P-MFI-calred	21000	8000	200	10	13.7	21.0	33.3	7.0	266.3	This work
38	Rh-MFI-cal	2100	800	120	10	6.6	86.7	-	-	-	This work
14	Rh-V/SiO ₂	3600	3600	115	1	0.2	7	0.098	0.0069	0.24	11
39	Rh/SiO ₂	6000	-	180	10	4	45	2.5	1.1	42	9
11	0.6Rh0.23Co/SiO ₂	6000	-	180	10	~9	52	5.5	2.9	42	9
40	Rh-0.7W/Al ₂ O ₃	6428	-	100	10	12	94	10.5	9.9	-	14
41		2142	-	100	10	27	96	7.9	7.6	-	14
30	RhCo ₃ /MCM-41	3600	-	160	1	17.7	60	6.5	3.9	~130	21
	Rh/2.9ReO _x -Al ₂ O ₃	46000	-	150	1	-	45	-	-	~3.6	10
	DPPPTS-RhAl ₁ /SiO ₂ ^b	-	2000	120	10	-	-	-	-	134	25
	Rh-Dpppe/SiO ₂ ^b	6000	-	140	10	-	97	-	-	1172	26

	Rh-Xantphos- SILP/NC ^b	7400	-	120	8	~70	95	■	■	600	27
	Rh/POL-PPh ₃ ^b	-	2000	120	10	96.2	96.1	■	■	4530	28
	Rh/POL-PPh ₃ ^b	-	5000	120	10	88.8	95.4	■	■	10373	28
	Rh ₂ P/SiO ₂ ^b	-	2773	200	50	53	75	■	■	~190	8

^a With a green background correspond to reactions at 200 °C.

^b With a brown background correspond to phosphine based catalysts.

Figure R3. Ethylene conversion and propanal selectivity relationship of all state-of-the-art-phosphine free solid catalysts. References and reaction conditions in Table S2 and S3.

Reviewer #4 (Remarks to the Author):

In the revised manuscript, the authors have been able to address all the concerns raised by this reviewer. The quality of the manuscript has been increased with all the additional experiments suggested by this reviewer and by the other reviewer. This reviewer recommends the acceptance of the manuscript in its present form.

Answers: We thanks the referee for his/her positive comments.

Reviewer #5 (Remarks to the Author):

The authors have clarified all my concerns and included in the revised manuscript. Even though the role of phosphate is not clearly understood as also other reviewers raised the same, future work in this line will certainly help in unravelling the catalyst structure. I recommend for publication.

Answers: We thanks the referee for his/her positive comments.

REVIEWERS' COMMENTS

Reviewer #3 (Remarks to the Author):

The authors have included additional experiments and discussions to address the concerns. As a result, the reviewer suggests acceptance in its current form.